Corrected: Author correction

# Atomic layer deposited Pt-Ru dual-metal dimers and identifying their active sites for hydrogen evolution reaction

Lei Zhang[1,8], Rutong Si[2,3,8], Hanshuo Liu[4,8], Ning Chen[5], Qi Wang[6], Keegan Adair[1], Zhiqiang Wang [7], Jiatang Chen [7], Zhongxin Song[1], Junjie Li[1], Mohammad Norouzi Banis [1], Ruying Li[1], Tsun-Kong Sham[7], Meng Gu[6], Li-Min Liu [3]*, Gianluigi A. Botton [4]* & Xueliang Sun[1]*

Single atom catalysts exhibit particularly high catalytic activities in contrast to regular nanomaterial-based catalysts. Until recently, research has been mostly focused on single atom catalysts, and it remains a great challenge to synthesize bimetallic dimer structures. Herein, we successfully prepare high-quality one-to-one A-B bimetallic dimer structures (Pt-Ru dimers) through an atomic layer deposition (ALD) process. The Pt-Ru dimers show much higher hydrogen evolution activity (more than 50 times) and excellent stability compared to commercial Pt/C catalysts. X-ray absorption spectroscopy indicates that the Pt-Ru dimers structure model contains one Pt-Ru bonding configuration. First principle calculations reveal that the Pt-Ru dimer generates a synergy effect by modulating the electronic structure, which results in the enhanced hydrogen evolution activity. This work paves the way for the rational design of bimetallic dimers with good activity and stability, which have a great potential to be applied in various catalytic reactions.

[1] Department of Mechanical and Materials Engineering, The University of Western Ontario, London, ON N6A 5B9, Canada. [2] Beijing Computational Science Research Center, Beijing 100193, China. [3] School of Physics, Beihang University, Beijing 100083, China. [4] Department of Materials Science and Engineering, McMaster University, Hamilton, ON L8S 4L8, Canada. [5] Canadian Light Source Inc, Saskatoon, SK S7N 2V3, Canada. [6] Department of Materials Science and Engineering, Southern University of Science and Technology, Shenzhen 518055, China. [7] Department of Chemistry, University of Western Ontario, London, ON N6A 5B7, Canada. [8] These authors contributed equally: Lei Zhang, Rutong Si, Hanshuo Liu. *email: liminliu@buaa.edu.cn; gbotton@mcmaster.ca; xsun@eng.uwo.ca

Single atom catalysts (SACs), a new frontier in catalysis with the highest atom utilization and abundant active sites, have shown particularly high catalytic activities compared to regular metal nanoparticles (NPs)[1–4]. By decreasing catalyst size to the single atom level, the atom utilization efficiency would be maximized[5–8]. Simultaneously, the electronic structure of catalysts is tuned by their coordination number, quantum size effects and the interaction with the support. Thus, SACs usually exhibit distinct catalytic properties as compared to metal NPs. Many methods have been developed for preparing different types of SACs, such as co-precipitation methods[9–12], wet impregnation methods[13–16], atomic layer deposition (ALD) methods[17–21], spatial confinement strategies[22,23–28], and photochemical reduction methods[29–31], etc. Recently, dual-metal sites, such as Fe–Co and Co–Zn, were found exhibit greatly enhanced activities, which was attributable to the synergistic effects between the electronic structures of the two elements[32,33]. However, the dual-metal sites were all synthesized by pyrolyzing two metal precursors simultaneously, which can not precisely control the location of each metal site. Therefore, it remains a great challenge to synthesize high-quality one-to-one A-B bimetallic dimer structures.

Among different methods for the preparation of SACs, ALD method can provide a precise control of the catalysts from the single atoms to nanoclusters, which makes it a powerful tool for investigating the relationship between the SAC structure and their catalytic performance. In 2013, our group firstly reported the synthesis of atomic layer deposited Pt single atoms on graphene nanosheets[17]. The as-prepared Pt single atoms exhibit greatly enhanced electrochemical activities in contrast to commercial Pt/C catalysts. Besides, the ALD technique provides the possibility to achieve atomically precise ultrafine metal clusters, even bimetallic sites. It should be pointed out that the ALD prepared dimer structures might be distinct from the reported dual-metal-sites. Through careful control of the deposition conditions, the second metal (e.g., Ru) can only attach on the preliminary one (e.g., Pt), selectively forming the bimetallic dimer structure.

Pt–Ru bimetallic catalysts have been proven to be an effective catalyst for several electrochemical catalytic reactions[34,35]. It can be expected that the electronic structure of Pt–Ru will be significantly tuned with the formation of bimetallic Pt–Ru dimer structure. Herein, we report an ALD route to prepare Pt–Ru dimers on nitrogen-doped carbon nanotubes (NCNT). The as-prepared Pt–Ru dimers show much higher mass activity and significantly improved stability compared with commercial Pt/C catalysts for hydrogen evolution reaction (HER). The detailed structure of the Pt–Ru dimers has been investigated by scanning transmission electron microscopy (STEM), X-ray absorption near edge structure (XANES), and extended X-ray absorption fine spectra (EXAFS). Density functional theory (DFT) calculation results indicate that the Pt atom strongly affects the electronic structure of the Ru atom, where the bimetallic dimer proceeds metallicity to covalence transformation, which results in the high HER performance.

## Results

### ALD preparation and characterization of Pt–Ru dimers.
As shown in Fig. 1, we prepared the Pt–Ru dimers on NCNTs via a two-step ALD process. Firstly, the isolated Pt atoms were obtained by using trimethyl(methylcyclopentadienyl)-platinum (IV) (MeCpPtMe₃) as the precursor, and nitrogen (99.9995%) as purge gas. During the ALD process, the Pt precursor tends to absorption and react with the N atoms on NCNTs, forming the strong metal-support interaction via the chemical bonding between Pt SAs and N-doping sites. The Pt single atoms were characterized by aberration-corrected high angle annular dark field scanning transmission electron microscopy (HAADF-STEM) imaging. The contrast in HAADF-STEM images is highly dependent on the atomic number (Z) of the elements in the material and linearly related the thickness of the sample. Therefore, compared to the carbon support, the heavier Pt atoms exhibited brighter intensity in the HAADF-STEM images. As shown in Supplementary Fig. 1 that many isolated Pt atoms (sharp bright spots) are uniformly dispersed on the NCNTs. The Pt loading amount was confirmed to be 0.9 wt% by inductively coupled plasma-optical emission spectrometer (ICP-OES). As the ALD temperature for Ru deposition is 270 °C, which is higher than that for Pt ALD, we maintained the Pt single atoms in ALD chamber at 270 °C for 1 h to investigate the stability of the Pt single atoms. As shown in Supplementary Fig. 2, the Pt single atoms maintained their structure at the Ru ALD deposition conditions.

Then the Pt–Ru dimers were achieved by ALD of Ru on Pt single atoms using bis(ethylcyclopentadienyl)ruthenium(II) as precursors. Recently, we reported that Ru atoms can not be deposited on NCNTs in the first several ALD cycles[36]. This result indicates that Ru atoms are not effectively attached onto NCNTs during the several initial ALD cycles, which provided the necessary prerequisite for selectively deposition of Ru atoms on Pt atoms. The atomic resolution HAADF-STEM images illustrate that a dimer-like structure was successfully synthesized by the designed ALD process (Fig. 2a–c, and Supplementary Fig. 3). The two atoms in the dimer show different contrast (Fig. 2d), which indicates the dimer-like structure is composed of two different elements (in this case are Pt and Ru atoms). In addition, the appearance of the Pt–Ru dimer structure is further confirmed by the X-ray absorption spectroscopy (XAS) results that discussed below. Although the dimer structure is not very uniform around the sample, the ratio of dimer structures is around 70% (Fig. 2e), indicating the significant amount of this type of structures in the material prepared. The Pt–Ru bonding of the dimers was statistically analysed and showed a distance of 0.24 ± 0.04 nm (Fig. 2f). The influence of the ALD temperature for the deposition of Pt and Ru was also investigated. With a decrease in the Pt ALD process temperature from 250 °C to 200 °C and 150 °C, ICP-OES results indicated that the Pt loading amounts on NCNT reduced from 0.9 to 0.65 and 0.42 wt%, respectively. Upon deposition of Ru on these three samples with different Pt loading at 270 °C, the Ru loadings was systematically reduced from 0.31% to 0.23 and 0.14 wt% (Fig. 2g). These results also directly demonstrate the selective deposition of Ru on Pt and the formation of dimer structures.

### X-ray absorption fine structure of Pt–Ru dimers.
We carried out XAS measurements to further investigate the electronic environment of Pt and Ru atoms in dimers. Figure 3a, b shows the normalized XANES spectra at Pt $L_3$ and $L_2$ edge, respectively. By qualitative and quantitative analysis of the Pt $L_2$ and $L_3$ edges white lines (WLs), the detailed Pt d state electronic structure can be revealed[37]. It is apparent that Pt foil exhibits a strong $L_3$-edge WL and a very weak $L_2$-edge WL due to the large spin orbit coupling of the 5d and an even distribution of the $5d_{5/2}$ and $5d_{3/2}$ densities of states just above the fermi level in metallic Pt; while both the dimer and single atoms exhibit substantial WL intensity at Pt $L_3$ and $L_2$ edges. In addition, the Pt–Ru dimers have the most intense WL among the three spectra. Note that the integrated area under the WL peak of Pt $L_{2,3}$-edge can directly reflect the unoccupied density of states of the Pt 5d orbitals. With the increase in the $L_{2,3}$-edge WL intensity, the number of electrons in the occupied d band decreases. In addition, the first derivative of the XANES spectrum shows the threshold energy $E_0$ for the Pt

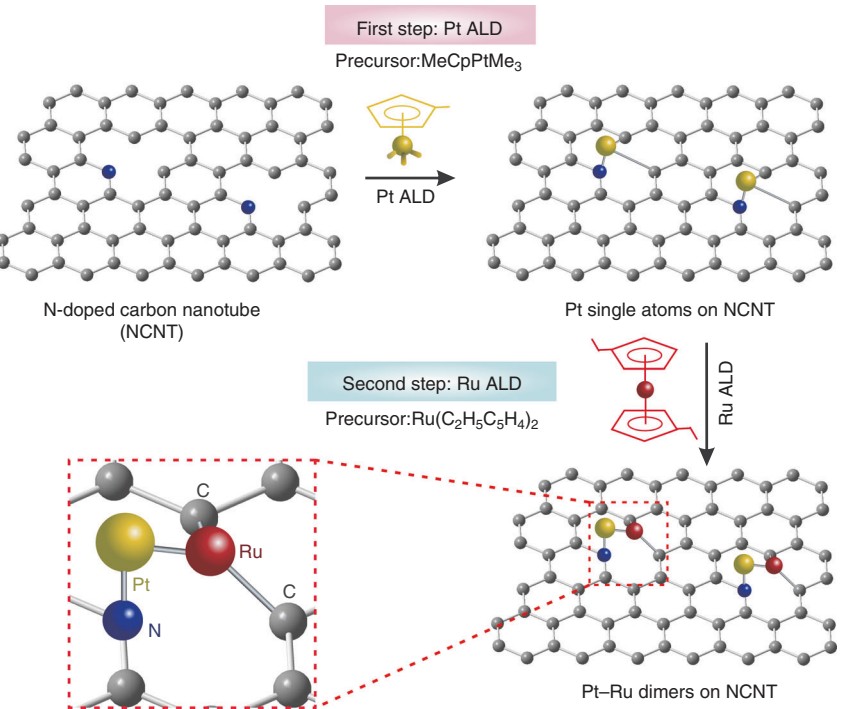

**Fig. 1** Schematic illustration of ALD synthesis of Pt–Ru dimers on nitrogen-doped carbon nanotubes (NCNTs). Firstly, the Pt single atoms were deposited by using MeCpPtMe₃ as the precursor. Then the Pt–Ru dimers were prepared by selective deposition of Ru atoms on Pt single atoms. Gray: C, Blue: N, yellow: Pt, red: Ru

single atoms, Pt–Ru dimers and Pt foil (Supplementary Fig. 4). Among the three samples, the $E_0$ for Pt single atoms and Pt–Ru dimer structure are 11565.5 and 11565.9 eV, which are much higher than that of 11564.0 eV of Pt foil. This result means that Pt is depleted in d charge. To further explore the implication of the electronic information of 5d states in Pt, we quantitatively analyzed the occupancy of the Pt 5d states in each sample based on a reported method[38]. Table 1 summarizes the detailed information of Pt L₃- and Pt L₂-edge threshold. From the analysis, the Pt–Ru dimers have the highest total unoccupied density of states of Pt 5d character (1.09), while the Pt 5d character of Pt foil is only 0.70. Several previous studies showed that the high unoccupied density of Pt d-orbitals played an important role in enhancing the activity of catalysts[17,18,39].

In addition to XANES spectra, the EXAFS spectrum was analyzed to investigate the local structure environment of Pt. Figure 3c shows the magnitude of Fourier transforms (FT) of the Pt EXAFS for Pt single atoms, Pt–Ru dimers and Pt foil. The EXAFS R space curve attributes the FT magnitude peak at 1.6 Å to Pt–C/N/O upon which Pt is anchored. To exclude the Pt–O bond, we tested the XANES and EXAFS spectrum of PtO₂ (Supplementary Fig. 5). As shown in the FT spectra from EXAFS of the PtO₂, two obvious peaks can be observed at around 2.7 and 3.3 Å, respectively. However, in the Pt single atom sample, we did not observe a peak at 2.7 either 3.3 Å (Fig. 3c). This result indicates that no Pt–O formed in Pt single atoms. In addition, previous studies showed that the metal single atoms can be stabilized by bonding with C and N on the substrates.[40–43] Thus, we concluded that the peak at 1.6 Å is attributed to Pt–N/Pt–C bond. The peak at 2.6 Å can be ascribed to the Pt–Pt or Pt–Ru bonding. For the Pt foil, the first shell FT peak is resolved at 2.6 Å, corresponding to Pt–Pt bonding. For the Pt–Ru dimers, in addition to an obvious Pt–N/Pt–C peak, a relative weak feature resolved at around 2.6 Å. As the dimer structure is prepared from Pt single atoms, the peak at 2.6 Å is arising from the Pt–Ru bonding.

XAS was also used to investigate the Ru electronic structure change in the Pt–Ru dimer catalysts (Fig. 3d). The XANES Ru K edge of Pt–Ru dimer sample shows a relatively broader curve features compared to that of Ru metal. The Ru K-edge of the dimer does not exhibit the beating of oscillations characteristic of the hexagonal Ru metal. A simple boarded sinusoidal oscillation indicates that there is no long range order with the presence of a Pt–Ru pair plus substrate low z atoms. In addition, a higher energy shift of the edge for Pt–Ru dimers compared to Ru metal can be observed from the Ru K XANES spectra, which indicates the charge redistribution also in favor of a charge depletion at the Ru site relative to Ru metal. The positive $E_0$ shift of Ru is due to the bonding between Ru atoms and C on the substrates. The fact that both Pt L₃-edge and Ru K-edge show a small threshold shift is due to the combined interaction of the Pt–Ru dimers with the substrate. The FT of the EXAFS region for Pt–Ru dimers and Ru foil are shown in Fig. 3e. The Pt–Ru dimers exhibited an obvious Ru–C peak at around 1.6 Å and a relative weak peak at around 2.8 Å. The peak below 2 Å can be assigned to bonds formed between Ru and C. The small peak at 2.8 Å is attributed to the Pt–Ru scattering by R space curve fitting (see more detailed analysis below). It should noted that since the k dependence of the backscattering amplitude in these heavy atoms exhibit oscillations in k space, the FT of the EXAFS is no longer a simple symmetric peak as in low z scattering atoms like C or N where the k dependence of f(k, π) is monotonic.

**Electrocatalytic performance of Pt–Ru dimers towards HER.** Cyclic voltammograms (CV) of the Pt–Ru dimers, Pt single atoms and commercial Pt/C were recorded in 0.5 M H₂SO₄ at a scanning rate of 50 mV s⁻¹. As the Pt loading of Pt–Ru dimers and Pt single atoms is extremely low, no obvious hydrogen adsorption/desorption peak is observed (Supplementary Fig. 6). As Ru in the dimer structure can adsorb and desorb O more easily through an oxidation–reduction process than Pt, the CV curve in

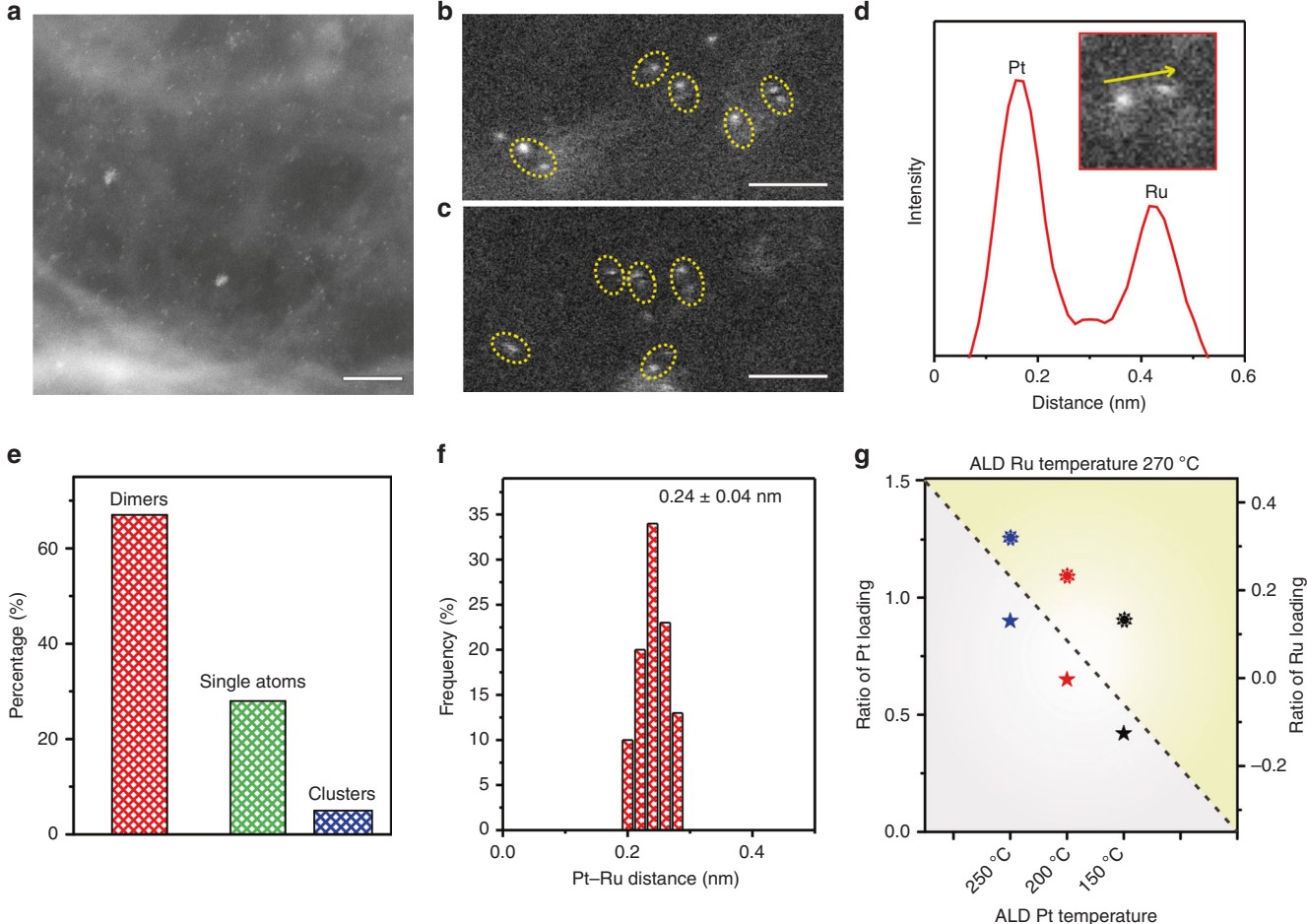

**Fig. 2** Characterizations of Pt–Ru dimers. **a–c** Aberration-corrected HAADF-STEM images of Pt–Ru dimers/NCNTs. Scale bars, 5 nm (**a**); 1 nm (**b**, **c**). The slowly varying contrast is due to the changes in the thickness of the substrate. **d** The intensity profile obtained on one individual Pt–Ru dimer. **e** Distribution histogram showing the ratio between dimers, single atoms and clusters. We determined the percentage of dimers by counting 200 clusters. **f** Pt–Ru distance in the observed Pt–Ru dimers. **g** The ratio of Pt and Ru loading in Pt–Ru dimers/NCNTs, which were obtained under different ALD conditions

Supplementary Fig. 6b exhibited a weak O adsorption/desorption peak. Figure 4a shows the HER curves of the catalysts, which were measured by conducting linear sweep voltammetry measurements in 0.5 M $H_2SO_4$ at room temperature. To directly compare the activity of Pt–Ru dimers, Pt single atoms and commercial Pt/C catalysts, we normalized the current density with the geometric area of the electrode to obtain the specific activity for each catalyst (Supplementary Fig. 7). The result indicates that both the Pt–Ru dimers and Pt single atoms exhibit comparable specific activity with Pt/C. In addition, the Pt–Ru dimers exhibit better HER activity than Pt single atoms. We further calculated the mass HER activities based on the mass loading at the overpotential of 0.05 V (The metal mass loadings for Pt/C, Pt single atoms, and Pt–Ru dimers on the electrode are 61.2, 1.34 and 1.67 $\mu g\ cm^{-2}$, respectively). The Pt–Ru dimers shows the mass activity of 23.1 A mg$^{-1}$ at the overpotential of 0.05 V, which is 54 fold greater than the Pt/C catalysts (0.43 A mg$^{-1}$). Furthermore, the mass activity of Pt–Ru dimers exceeds most of the other state-of-the-art Pt-based catalysts (Supplementary Table 1). These results indicate that the Pt–Ru dimer structures can effectively improve the HER activity in contrast to single atoms and NPs. We also employed Tafel plots to illustrate the HER kinetics of the catalysts. As shown in Supplementary Fig. 8, the Tafel slope of Pt–Ru dimers, Pt single atoms and Pt/C catalysts was 28.9, 33.2 and 29.7 mV dec$^{-1}$, respectively. Empirically, the Tafel slope value can attributed to

the slow atomic combination to form $H_2$. It should be noted that the Pt SA is different from the Pt surface, and the former could adsorb more than one hydrogen atom on the one isolated atom, and each Pt atom of the later typically adsorbs one hydrogen atom at the surface[18,30,31]. Thus the Pt SA can process the HER process on the isolated Pt atom through the Tafel half reaction ($H_{ads} + H_{ads} \rightarrow H_2$).

To evaluate the durability of the Pt–Ru dimers and Pt SACs, accelerated degradation tests (ADTs) were adopted between +0.4 and −0.15 V (versus RHE) at 100 mV s$^{-1}$ for 5000 cyclic voltammetry sweeps. As shown in Fig. 4c, the polarization curve of Pt–Ru dimers after ADT tests exhibited a very similar activity as the initial test. The HER activity at the overpotential of 0.05 V only showed a 5% loss for the Pt–Ru dimers (Fig. 4d). While, the Pt SACs showed a 9% loss at the overpotential of 0.05 V after ADT test. The Pt/C catalysts exhibited the worst stability, and the activity dropped 28% after 5000 cycles at an overpotential of 0.05 V (Supplementary Figs. 9–11). In addition, the TEM images of the post-testing Pt–Ru dimer catalysts indicated that the dimer structure is stable after the durability test (Supplementary Fig. 12). The different intense of two atoms in one dimer indicate the existence of dimer structure. We also carried out the XAS test for the dimer structures after HER test. The XANES spectrum of Pt has almost no change. In the K2-weighted magnitude of FT spectra from EXAFS spectrum, a small peak at 2.8 Å attributed to

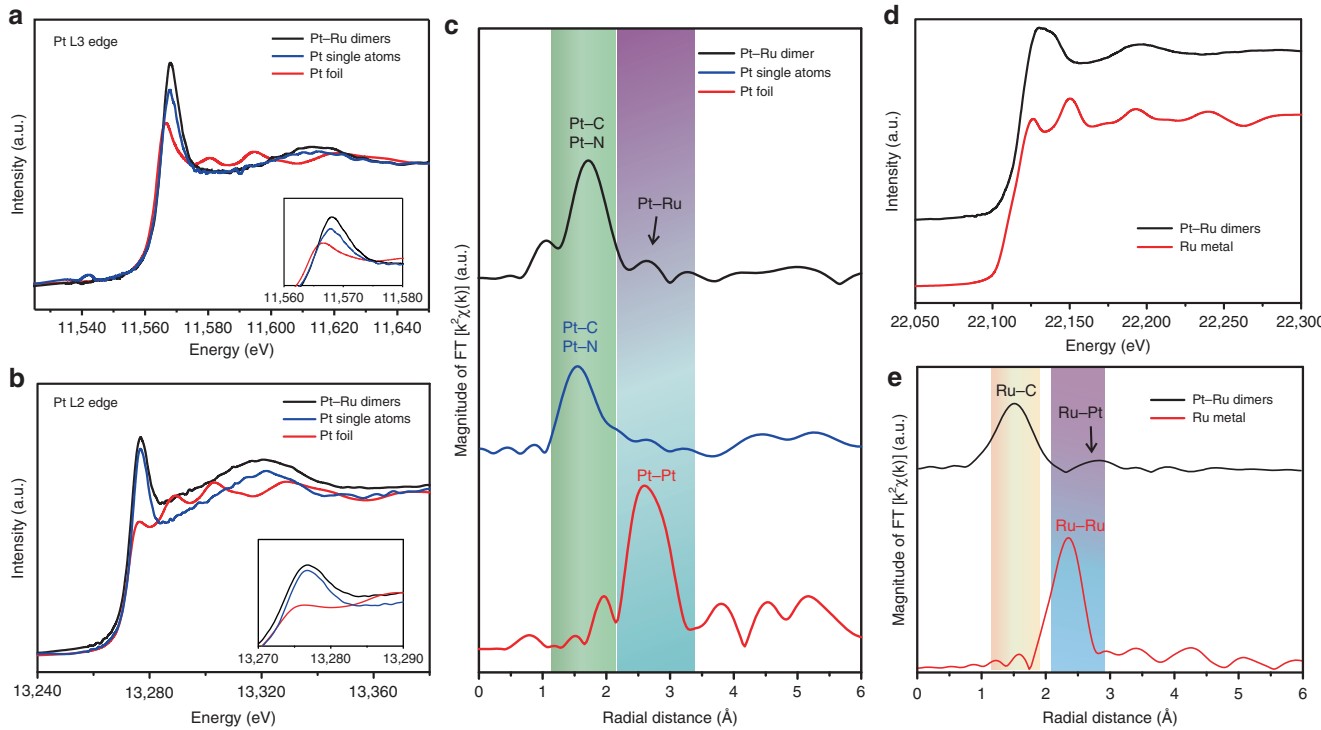

**Fig. 3** X-ray absorption studies of the Pt–Ru dimers, Pt single atoms and Pt foil. **a, b** The normalized XANES spectra at the Pt L3-edge and L2-edge of the Pt–Ru dimers, Pt single atoms and Pt foil. **c** corresponding K2-weighted magnitude of Fourier transform spectra from EXAFS of the Pt–Ru dimers, Pt single atoms and Pt foil. **d** The normalized XANES spectra at the Ru K-edge and **e** corresponding K2-weighted magnitude of Fourier transform spectra from EXAFS of the Pt–Ru dimers and Ru metal

| Table 1 Pt $L_{3,2}$-edge threshold and white line (WL) parameters | | | | | | | | |
|---|---|---|---|---|---|---|---|---|
| Sample | Pt $L_3$ edge WL | | | Pt $L_2$ edge WL | | | $h_{5/2}$ | $h_{3/2}$ | Total |
| | $E_0$ (eV)[a] | $E_{Peak}$ (eV)[b] | $\Delta A_3$[c] | $E_0$ (eV) | $E_{Peak}$ (eV) | $\Delta A_3$ | | | |
| Pt foil | 11564 | 11566.6 | 6.36 | 13273 | 13276.5 | 4.1 | 0.51 | 0.19 | 0.70 |
| Pt single atom catalysts | 11565.5 | 11567.7 | 7.42 | 13273.7 | 13276.6 | 5.9 | 0.67 | 0.27 | 0.94 |
| Pt–Ru dimers | 11565.9 | 11567.9 | 8.13 | 13273.6 | 13276.7 | 7.9 | 0.72 | 0.37 | 1.09 |

[a]Position of the first inflection point of the edge jump for the corresponding Pt $L_3$ edge
[b]Peak position
[c]Area under the difference curve for normalized edge jump, the normalized edge jump for the Pt $L_3$ and $L_2$ edge corresponds to a value of $2.5 \times 10^3$ cm$^{-1}$ and $1.16 \times 10^3$ cm$^{-1}$, respectively

the Pt–Ru scattering can still be observed (Supplementary Fig. 13). For the Ru XAS spectra, three XANES features A, B and C can be reproduced, indicating the maintenance of the Pt–Ru dimer structure (Supplementary Fig. 14). We also carried out the long-term stability of the Pt–Ru dimers, Pt single atoms and commercial Pt/C by extended electrolysis at fixed current density of 10 mA cm$^{-2}$ for 10 h. Supplementary Fig. 15 shows that the Pt–Ru dimers and Pt single atoms exhibit much better HER stability than Pt/C catalysts. Compared to the 95.1 mV potential drop for the Pt/C, there are only 19.6 and 56.9 mV potential drop for Pt–Ru dimers and Pt single atoms, respectively. Thanks to the strong interaction between Pt atoms with the N-site, both Pt–Ru dimers and Pt single atoms show better stability than the Pt/C catalysts.

**Identifying the active sites of dimers by DFT calculations.** We performed DFT calculations to deeper understand the origin of the HER activity of Pt–Ru dimers. Considering the Ru atoms were deposited after the Pt SAs were formed, most of the N-sites

have been occupied by the Pt atoms. Thus the Ru atoms could only either bond with Pt to form Pt–Ru dimer or form Ru–Ru dimer. The adsorption energies of Ru–Ru dimer on the perfect site and Pt–Ru dimer on the N-doped site were calculated to be −1.3 and −4.87 eV, which indicates that the Ru prefers to form the Pt–Ru dimer instead of Ru–Ru dimer. In addition, as we clearly identify the Pt–Ru atomic pairs from HAADF-STEM images and Pt–Ru bonding from EXAFS spectra, the following first-principle calculations were carried out using one Pt–Ru dimer on the N-doped graphene to identify reactivity of the dimer. The EXAFS R space curve fitting and DFT model guided Ru K edge XANES theoretical modeling demonstrate the validity of the DFT models, which will be discussed below. We explored how the H adsorption evolves on the Pt–Ru dimer step by step, and the different numbers of H adsorption on the Pt–Ru dimer were examined. As for one hydrogen atom, it prefers to adsorb on the Ru atom at the adsorption energy of −3.3 eV rather than the Pt atom (−2.5 eV). It should be noted that the second H is not so easy to adsorb on the Pt–Ru dimer compared with the first one,

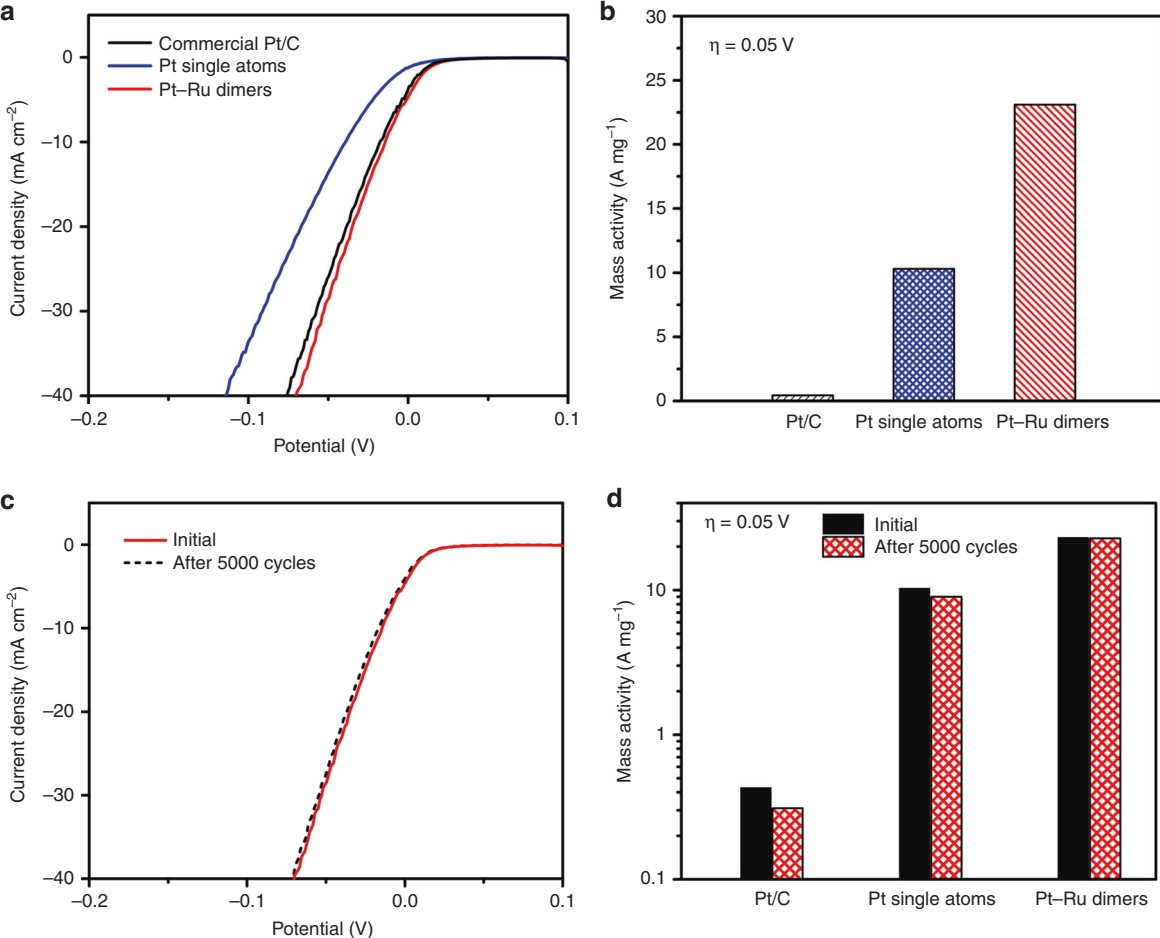

**Fig. 4** Electrocatalytic performance of Pt–Ru dimers towards HER. **a** The HER polarization curves recorded on Pt–Ru dimers, Pt single atoms and Pt/C catalysts. **b** Normalized mass activity at 0.05 V of Pt–Ru dimers, Pt single atoms and Pt/C catalysts. **c** Durability measurement of the Pt–Ru dimers catalysts. **d** Normalized mass activity at 0.05 V of Pt–Ru dimers, Pt single atoms and Pt/C catalysts before and after durability tests. The Pt mass loadings for Pt/C, Pt single atoms, and Pt–Ru dimers are 61.2, 1.34 and 1.24 µg cm$^{-2}$, respectively

while with the further increasing number of the hydrogen atoms, the H adsorption becomes favorable again, which should be related with the change of electronic structure for Pt–Ru dimer.

Different from the NPs[18], each Pt or Ru atom of dimer could adsorb more than one hydrogen atoms. Thus, we further studied the total hydrogen number of H adsorption on Pt–Ru, which was gradually increased from 2 to 6, and the results were shown in Fig. 5 and Supplementary Table 2. Interestingly, after the first step, the hydrogen atoms first adsorb on the Pt side one by one, after three hydrogen atoms adsorb on Pt side and one hydrogen on Ru side, the next one prefers to stay on the Ru atom in sequence. By repeating this process, the maximum number of adsorbed H atoms on either Pt and Ru atoms reach 3. In the following, we referred it as Pt(nH)Ru(mH). As shown in Fig. 5a, before the final step, two possible mediate structures, Pt(3H)Ru (2H) and Pt(2H)Ru(3H), could be formed, and the Pt(3H)Ru(2H) is the energetically more stable. The bond distance between Pt and Ru atoms increases from 2.38 Å (Pt(0H)Ru(0 H)) to 2.75 Å (Pt(2H)Ru(3H)) with the increase of the H adsorption, while it becomes 2.72 Å for the Pt(3H)Ru(2H), which is about 0.1 Å longer than the last step. To understand the stability of the pure Pt–Ru and Pt–Ru with 6H on the N-doped graphene, the first principles molecular dynamics (FPMD) were carried out for 5 ps with the target temperature of 300 K. The results show that both Pt–Ru dimer with and without the six H were stable, and no big

structure change occurred during this process, which further confirm the stability of the Pt–Ru dimer (Supplementary Fig. 16).

The calculated hydrogen adsorption Gibbs free energies ($\Delta G_H$) of the Pt–Ru dimer were performed to examine the activity of the HER for different number of H atoms, and the results for typical configurations are shown in Fig. 5b. When one hydrogen atom adsorbs on both Ru and Pt sides of the Pt–Ru dimer, the corresponding Gibbs free energy for Pt(1H)Ru(1H) is about −1 eV. When another hydrogen adsorbs on the dimer, the corresponding Gibbs free energy for Pt(2H)Ru(1H) becomes about 0.6 eV. This means that the hydrogen atoms are anchored on Ru atom steadily or hard to detach from either Ru or Pt atom at this stage. As for the last step (see Fig. 5), three hydrogen atoms connect with the Pt atom (3H-Pt), and three bonds with Ru atom (3H-Ru). The Ru atom exhibits the low $\Delta G_H$ of 0.01 eV through the pathway of Pt (3H)Ru(3H) → Pt(3H)Ru(2H), which clearly indicates that the hydrogen atom becomes easy to detach from Ru when the maximum coverage of 6H is reached. The corresponding $\Delta G_H$ is 0.01 eV, which is even smaller than that of the single atom Pt (see Supplementary Fig. 17 and Supplementary Table 3)[44]. Meanwhile, the calculated $\Delta G_H$ of Pt–Pt dimer is about −0.14 eV (Supplementary Fig. 18), which is quite close to that the bulk one, inferior to that of Pt–Ru as well. This suggests that the Ru of Pt–Ru dimer should play the vital role in the HER reaction although Pt NPs generally exhibit high activity for HER.

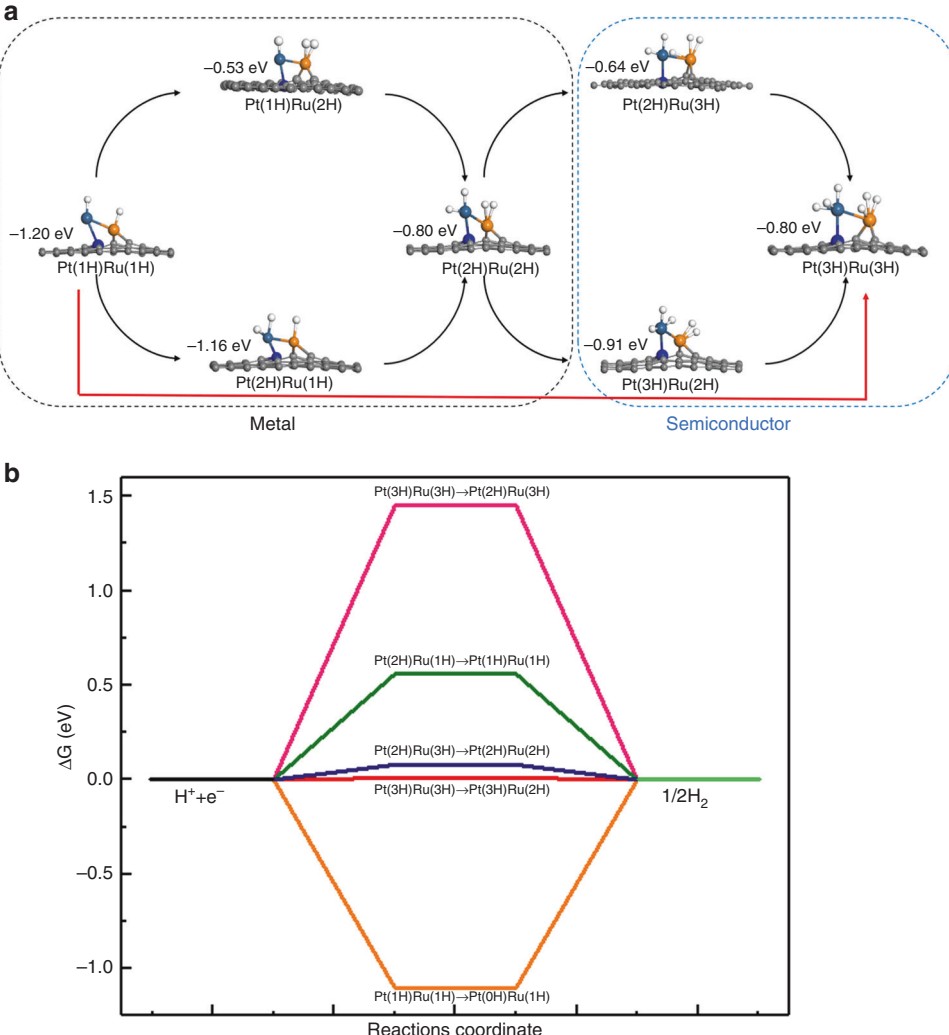

**Fig. 5** Hydrogen adsorption configurations on the Pt–Ru dimer and the $\Delta G_H$ for different coverages of typical configurations. **a** The atom structures of different H adsorption along with the adsorption energies. **b** The schematic of $\Delta G_H$ for the different hydrogen coverages of typical configurations. For each coverage, all the possible pathways were considered, and the lowest $\Delta G_H$ for each coverage was typically shown, except we noted. Here, Pt(nH)Ru(mH) represents n H atoms adsorbed on Pt and m H atoms on Ru. Orange, dark blue, light blue and gray balls represent the Ru, Pt, N and C atoms, respectively

In order to further understand why the dimer exhibits the superior activity, the detailed electronics structures for the different stages are explored. The partially density of states (PDOS) for both the Pt and Ru atoms are examined, and the results are shown in Fig. 6a and Supplementary Figs. 19–22. As shown in Supplementary Fig. 19, before the hydrogen adsorption, both Pt atom and Ru atom of dimer exhibits unoccupied states, which should origin from the bonding between metal ion and C/ N atoms. Meanwhile such results also agree well with our XAS experiments as discussed above. With the increase of the hydrogen adsorption, the calculated PDOS also suggested that both the Pt and Ru gradually lost the metallic, resulting more unoccupied states, when the two hydrogen atoms adsorb on both Pt and Ru (Pt(2H)-Ru(2H)), respectively (see Supplementary Fig. 20 and Fig. 5a). As shown in Fig. 5a and Supplementary Table 2, the $\Delta G_H$ becomes small after the electronic structures of both Pt and Ru become semiconductor. Meanwhile with the increase of the hydrogen adsorption, the $\Delta G_H$ is further reduced, thus the electronic structure of Pt–Ru dimer under the hydrogen adsorption plays a vital role in modulating the HER activity of Pt–Ru dimer.

Considering the Pt–Ru atom show the small $\Delta G_H$ under the high hydrogen coverage (Pt(3H)Ru(3H)), the corresponding oxidation states for Pt and Ru are explored based on the occupation number and PDOS. Before Pt–Ru dimer reaches the best HER activity, the Ru atom interacts with two C atoms of the substrate and two hydrogen atoms and the Pt bonds with one N atoms and three hydrogen atoms (Pt(3H)Ru(2H)). The corresponding crystal field of the Pt and Ru are octahedral and tetrahedral (see Fig. 6b–d), respectively. In this situation, the d$xy$, d$z^2$, d$x^2-y^2$, and d$xz$ are half-filled, and the d$yz$ orbital is fully unoccupied, thus the Ru atom corresponds to 4+ for Pt(3H)Ru (2H). As for the Pt, the d$x^2-y^2$ is half-filled, d$z^2$ is fully unoccupied and all other orbital are occupied, thus the Pt is 3+. In this situation, the hydrogen atom on Ru owning the smaller $\Delta G_H$ mainly stays in the direction of the d$xz$.

With the further H adsorption on the Ru atom from 2 to 3, the Ru is oxidized from 4+ to 5+ and Pt is reduced from 3+ to 2+. In this process, the bond distance between the Pt–Ru dimer decreases from 2.72 to 2.63 Å, which results in the electron redistribution between Pt–Ru dimer. The d$x^2-y^2$ of Pt becomes fully-occupied, and the Pt is reduced to 2+. Meanwhile, the bond

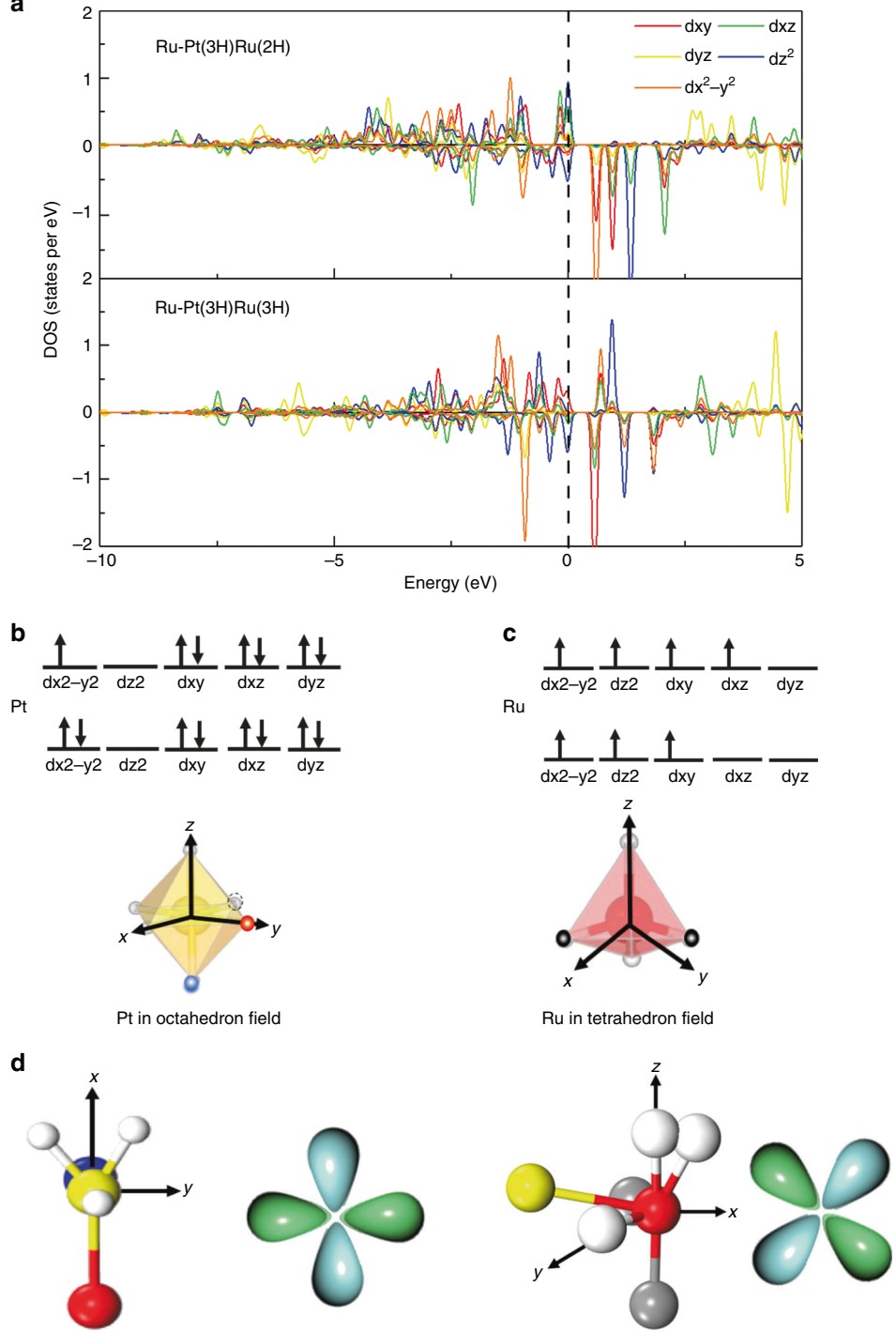

**Fig. 6** Electronic structures and crystal field of the Ru atom for the last two steps. **a** The Projected DOS of Ru atom for Pt(3H)Ru(2H) and Pt(3H)Ru(3H). **b** Schematic of crystal field of Ru. **c** The dyz orbital of Ru of Pt(3H)Ru(2H) and the dyz orbital of Ru of Pt(3H)Ru(3H). Here, Pt(nH)Ru(mH) represents n H atoms adsorbed on Pt and m H atoms on Ru. Red, yellow, blue and gray ball represent the Ru, Pt, N and C atoms, respectively

distance between the Ru with the C atoms of substructure decreases by 0.05 Å. The corresponding bonding between the Ru–C is dyz, thus the antibonding state of dyz is pushed to a rather high energy level (see Fig. 6a). Although $dx^2-y^2$, $dz^2$ and dxy of Ru remain half-filled, dxz changes from half-filled to fully unoccupied, and the antibonding orbital of dxz is greatly shifted to a rather high energy level. The hydrogen atoms on the Ru atom, exhibiting the smallest $\Delta G_H$, just interacts with the dxz orbital for both Pt(3H)Ru(2H) and Pt(3H)Ru(3H) (Supplementary Figs. 21 and 22 and Supplementary Table 4). Considering

that dxz orbital of Ru becomes unoccupied by the synergetic effect of Pt–Ru dimer during the hydrogen adsorption, the interaction between the hydrogen and the dxz of Ru atom should be rather weak, which should be the main reason why the Pt–Ru dimer exhibits the superior HER as found in our experiment.

## Discussion

To investigate the structure consistence between the catalysts and DFT models, the Feff (a software program used in XAS)

theoretical modeling is carried out. Firstly, we investigated the consistency between the Pt single atom DFT model and EXAFS theoretical modeling. We developed 3-shell R space curve fitting model base on the Pt single atom DFT model. Two shells are corresponding to Pt–C (coordination numbers, CN = 2) and Pt–N (CN = 1), which construct the triangle base of the single Pt atom tetrahedron configuration. The other shell is C shell (×6), which works as connection from the single Pt atom tetrahedron configuration to the substrate. The Pt local structural environment revealed by Pt L3 edge R space curve fitting result consists what predicted by DFT in terms of single Pt atom tetrahedron configuration and its imbedded connection with the N-doped carbon substrate Supplementary Fig. 23. As shown in Table 2, the Pt single atoms have the CN of 2.2 and 1.2 for Pt–C and Pt–N bonding respectively, which agrees well with our proposed DFT model.

For the dimer structure, we carried out the R space curve fitting of the Pt–Ru dimer structure guided on the crystallography predicted DFT model. 3-single-scattering path models Pt–N (CN = 1), Pt–Ru (CN = 1), and Pt–C (CN = 6) were used for the fitting. As shown in Supplementary Fig. 24, the overall agreement between the Feff fitting and the experimental data is revealed, suggesting that the obtained Pt–Ru dimer catalysts supports the modeling predicted DFT model structure. In addition to R space fitting, we also used XANES linear combination fitting (LCF) to identify the existence of Pt–Ru dimers in the catalysts. Because no single phase Pt–Ru dimer catalysts can be prepared, a corresponding experimental standard spectrum is not available for LCF analysis. The DFT guided theoretical XANES spectrum was used as a replacement in LCF. Three standard spectra, including Pt single atoms, Pt–Ru dimer and metallic Pt, were used for LCF analysis. As shown in Supplementary Fig. 25, LCF results a good mathematical fit. The result show that the percentage of Pt–Ru dimer and Pt single atoms are 15% and 85%, respectively. Because there are some Pt clusters in this sample, the amount of Pt single atom is overestimated. However, this data can prove the existence of certain amount Pt–Ru dimer in the sample. The above XAS fitting results (EXAFS R space curve fitting and XANES LCF analysis) strongly support the existence of Pt–Ru dimer and the rationality of the DFT predicted model.

We also developed Ru centered spherical cluster system based on the Pt–Ru dimer DFT model (Supplementary Fig. 26a). The structural system is composed by multiple Ru centered clusters, starting from 2.4 Å, 3.0 Å, 4.0 Å, to 5.0 Å. As shown in Supplementary Fig. 26b, comparison is made between experimental XANES spectrum and the best-fit of Ru XANES modeling (Ru centered clusters with the radius of 5.0 Å). The spectrum of Ru XANES modeling with the Ru radius of 5.0 Å shows a relatively strong B feature and a shoulder peak for A feature. It should be pointed that the peak intense and peak position of the fitted data might be a little different with the experimental data, because the model might have some defects compared with the actual Ru local structural environment of the sample. The presence of the three XANES features A, B and C can be used as the evidence of the formation of Pt–Ru dimers. Guided by the Pt–Ru dimer DFT model, an initial Feff modeling was performed based on the Ru centered cluster R5.0 Å. The modeled $k^2\chi(k)$ is compared to raw spectrum (Supplementary Fig. 27a), revealing a good reproducing of experimental data. The comparison was also made for the backward FT filtered $k^2\chi(k)$ (0.9–3.3 Å, latched region for the 1st two FT peaks) and the modeled result (Supplementary Fig. 27b). Both comparisons reveal that the experimentally resolved $k^2\chi(k)$ feature has been reproduced, supporting the reliability of the predicted Pt–Ru dimer DFT model. Based on the Pt–Ru DFT model, 5-path structure model was developed for R space curve fitting. The fitted parameters are summarized in Table 3. As

**Table 2 R space fitting results from Pt L₃ edge for Pt single atom catalysts and Pt–Ru dimers**

| Sample | Path | Coordination number | Bond length (Å) | $\sigma^2/10^{-3}$ Å² |
|---|---|---|---|---|
| Pt single atom catalysts | Pt–N | 1.2 (0.2) | 1.93 (0.02) | 2.1 |
| | Pt–C (1) | 2.2 (0.4) | 2.04 (0.02) | 1.6 |
| | Pt–C (2) | 5.5 (0.8) | 2.96 (0.05) | 9.4 |
| Pt–Ru dimers | Pt–N | 1.2 (0.3) | 2.06 (0.02) | 2.1 |
| | Pt–C | 5.2 (0.8) | 2.10 (0.02) | 8.4 |
| | Pt–Ru | 1.2 (0.2) | 2.34 (0.04) | 6.7 |

shown in Supplementary Fig. 28, the R space and K space curve fitting agrees well with the experimental data. From the Ru R space fitting results, the Ru atoms have the CN of 0.7 for Ru–Pt, also suggesting the formation of Pt–Ru dimer structure. It should be pointed out that R space curve fitting result (including CN and bond length) from Pt L3 edge and Ru K edge for the dimer Pt–Ru scattering path consists to each other. Namely, standing on Pt to see Ru is equivalent to standing on Ru to see Pt. Therefore, the spatial relation of Pt vs Ru consists the predicted dimer model by DFT.

In summary, we have successfully synthesized Pt–Ru dimers on NCNTs by the ALD method. The obtained Pt–Ru dimers showed significantly improved mass activity (more than 50 times) and excellent stability compared to commercial Pt/C catalysts for HER. Both XAS spectra and first-principles calculations indicate that the Pt–Ru dimers complex contains Pt–Ru bonding. Furthermore, the first-principles calculation reveals that the Pt–Ru dimer could be easily changed from metal to semiconductor by the adsorption, leaving unoccupied orbtials. The inertaction between H and Ru could be modulated by the Pt through the synergetic effect, which results in the high HER activity. This work provides an in-depth understanding of bimetallic dimer catalysis. Furthermore, we pave the way for the precise control of metal dimers, trimmers and even tetramers, which have great potential to be applied in various catalytic reactions.

## Methods

**Synthesis of NCNTs.** We synthesized the NCNTs with a diameter of 100 nm by ultrasonic spray pyrolysis according to a previous study[45]. In the synthesis process, we used imidazole as carbon and nitrogen source, and ferrocene as the catalyst precursor. The reaction were carried out at 850 °C. During the reaction, the ferrocene can decompose into iron particles, which acts as the catalyst for the growth of carbon nanotubes and the incorporation of nitrogen in the graphite layers to yield NCNTs. The received NCNTs were washed with nitric acid and water for 6 h at 80 °C, respectively. The NCNTs were loaded on aluminum foil before putting into the ALD reactor chamber.

**ALD synthesis of Pt–Ru dimers on NCNTs.** We used MeCpPtMe₃ as the precursor for the deposition of Pt single atoms onto the NCNTs. The Pt ALD process were carried out at 250 °C by ALD (Savannah 100, Cambridge Nanotechnology Inc., USA). The purging gas and carrier gas are both high-purity N₂ (99.9995%). The container for MeCpPtMe₃ was maintained at 65 °C for a steady-state flux of Pt to the chamber. Pt single atoms were obtained after a 30 s exposure of Pt precursor. After the formation of Pt single atoms on NCNTs, Ru atoms was selectively deposited on the Pt single atoms by using bis(ethylcyclopentadienyl)ruthenium(II) precursors. The carrier gas is high-purity N₂ (99.9995%). The container for bis (ethylcyclopentadienyl)ruthenium(II) was heated at 110 °C and the chamber temperature was kept at 270 °C. The gas lines were held at 150 °C. During the Ru ALD process, a 10 s exposure of Ru precursor was used for the deposition of Ru. Pt and Ru loading were analyzed using an inductively coupled plasma-optical emission spectrometer (ICP-OES) with samples dissolved in fresh hot aqua regia and filtered.

**HER activity measurements.** The electrochemical HER measurements were performed using a three electrode system. The glassy carbon rotating-disk electrode (Pine Instruments) was used as the working electrode. A graphite electrode was used as the counter electrode, and a reversible hydrogen electrode (RHE) was used

**Table 3 R space fitting results from Ru K edge for Pt–Ru dimers**

| Sample | DFT | | | R space curve fitting | | |
|---|---|---|---|---|---|---|
| | Path | Coordination number | Bond length (Å) | Coordination number | Bond length (Å) | $\sigma^2/10^{-3}$ Å$^2$ |
| Pt–Ru dimers | Ru-C | 2 | 1.90 | 2.4 (0.4) | 1.76 (0.02) | 2.9 |
| | Ru-Pt | 1 | 2.38 | 0.7 (0.1) | 2.36 (0.03) | 7.0 |
| | Ru-C | 3 | 2.58 | 2.5 (0.4) | 2.25 (0.03) | 7.0 |
| | Ru-N | 1 | 2.87 | 1.4 (0.2) | 2.83 (0.05) | 2.8 |
| | Ru-C | Multiple | $\geq 3.10$ | 3.9 (0.5) | 3.10 (0.06) | 7.1 |

as the reference electrode. We prepared the ink for the electrochemical measurement by mixing 3 mg of the catalysts, 1.6 mL DI water, 0.4 mL isopropanol, and 40 μL Nafion (5% solution, Sigma-Aldrich), followed by sonication for 10 min. Twenty microliters of the ink was loaded onto the glassy carbon rotating dis electrode (0.196 cm$^2$) for preparing the working electrode. The CVs were recorded in a N$_2$-saturated 0.5 M H$_2$SO$_4$ by cycling between 0.05 and 1.1 V$_{RHE}$ at a scan rate of 0.05 V s$^{-1}$. The HER test was carried out with a scan rate of 2 mV s$^{-1}$. To avoid the formation of H$_2$ gas bubbles at the catalyst surface during the HER tests, the working electrode was rotated at 1600 r.p.m.

**Instrumentation.** Transmission electron microscopy (TEM) samples were prepared by drop casting an ultrasonicated solution of the samples of interest dissolved in high-performance liquid chromatography grade methanol solution onto lacey carbon grids. The samples were tested on a FEI Titan Cubed 80–300 kV microscope equipped with spherical aberration correctors (for probe and image forming lenses) at 200 kV. The STEM images were collected with a high angle-annular dark field detector (HAADF) with a collection angle of ~64–200 mrad.

**X-ray absorption spectroscopy.** We used the beamline 20-BM-B at the Advanced Photon Source (APS, Argonne National Laboratory) and the hard X-ray micro-analysis (HXMA) beamline (Canadian Light Source) to obtain the Pt L$_2$, L$_3$-edge and Ru K-edge XAS spectra. The fluorescence yield mode with a solid-state detector was used to collect each spectrum. The spectra of Pt foil and Ru metal were collected in transmission mode for comparison and monochromatic energy calibration. The Athena software was used to analyze the obtained XAS data. The extracted EXAFS data was weighted by k$^3$, then converted to R space by FT to obtain the magnitude plots of the EXAFS spectra. The EXAFS data R space curve fitting was performed by WinXAS version 2.3[46].

**Quantitatively intensity analysis of XAS.** The Pt L$_{3,2}$-edge WL intensity was analyzed using the Au metal L$_{3,2}$-edge X-ray absorption near edge spectroscopy (XANES) as the background. The areas under the difference curve in the L$_3$ and L$_2$ WL region between Pt and Au were denoted as A$_3$ and A$_2$ at the respectively. The area under the difference curve was integrated between the two vertical bars. The ΔA$_3$ and ΔA$_2$ were obtained according to the following equations:

$$\Delta A_3 = \int (Pt)_{L3WL} - \mu(Au)_{L3WL} \qquad (1)$$

$$\Delta A_2 = \int (Pt)_{L2WL - \mu(Au)_{L2WL}} \qquad (2)$$

and according to Sham et al.[38], the ΔA$_3$ and ΔA$_2$ values were able to be calculated from the following equations:

$$\Delta A_3 = C_0 N_0 E_3 (R_d^{2p^2/3})^2 \left[\frac{6h_{\frac{5}{2}} + h_{\frac{3}{2}}}{15}\right] \qquad (3)$$

$$\Delta A_2 = C_0 N_0 E_2 (R_d^{2p^{1/2}})^2 \left(\frac{1}{3}h^{3/2}\right) \qquad (4)$$

where $C_0 = 4\pi\gamma^2\alpha/3$ (α is the fine structure constant), N$_0$ is the density of Pt atoms, and E$_2$ and E$_3$ are the E$_0$ for the L$_2$ and L$_3$ edges, respectively. The R is the radial transition matrix element, and h$_j$ is the 5d hole counts. By assuming R terms are similar for both edges

$$C = C_0 N_0 R^2 \qquad (5)$$

C value of $7.484 \times 10^4$ cm$^{-1}$ was derived previously for Pt metal.
and with this approximation

$$h_{3/2} = \left[\frac{3\Delta A_2}{C}\right] \qquad (6)$$

$$h_{5/2} = \frac{1}{2C}\left[5\frac{E_2}{E_3}\Delta A_3 - \Delta A_2\right] \qquad (7)$$

The calculated d hole counts (h$_{5/2}$ and h$_{3/2}$) are listed in Table 1.

**Computational method.** We performed first-principles calculations by the CP2K/Quickstep package[47]. The correlation energies and nonlocal exchange were described by Perdew–Burke–Ernzerhof functional[48]. The norm-conserving GTH pseudopotentials were used to describe the core electrons[49]. The m-DZVP were used for expanding the wave function of N 1s$^2$2s$^2$2p$^3$ and C 1s$^2$2s$^2$2p$^2$, and m-TZVP for Ru 4s$^2$4p$^6$4d$^7$5s$^1$ and Pt 4s$^2$4p$^6$5d$^9$6s$^1$ electrons[50]. The cut off energy of auxiliary basis set of plane waves is 500 Ry. During the calculations, all the atomic positions were fully relaxed until the force is smaller than 0.05 eV/Å. We performed FPMD by CP2K/Quickstep package.

The hydrogen adsorption energies, E$_{ad}$, were calculated according to the following equations,

$$E_{ad} = 1/n\left(E_{nH/metallic\ dimer} - E_{metallic\ dimer} - n/2\ EH_2\right) \qquad (8)$$

$E_{nH/metallic\ dimer}$ and $E_{metallic\ dimer}$ represent the total energies of n hydrogen atoms adsorbed on metallic dimer and metallic dimer, respectively. $E_{H2}$ is the total energy of one hydrogen molecule in vacuum, n refers to the number of hydrogen atoms adsorbed on metallic dimer.

The hydrogen-adsorption Gibbs free energies, ΔG$_H$, were calculated according to the following equations[44],

$$\Delta E_H = E_{(n+1)H/metal} - E_{nH/metal} - \frac{1}{2}E_{H2} \qquad (9)$$

$$\Delta G_H = \Delta E_H + \Delta E_{ZPE} - T\Delta S \qquad (10)$$

We defined the ΔE$_H$ as the hydrogen binding energy on metal atoms. $E_{(n+1)H/metal}$ and $E_{nH/metal}$ represent the total energies of $n + 1$ and $n$ hydrogen atoms adsorption on metal atom, respectively. $E_{H2}$ represent the total energy of one hydrogen molecule in vacuum. ΔE$_{ZPE}$ is the difference of the zero-point energy with and without hydrogen adsorption, T is 300 K, and ΔS is the entropy change between an adsorbed hydrogen and gas-phase hydrogen at 101,325 Pa.

## Data availability

All data are available from the authors, please refer to author contributions for specific data sets. Source data are provided as a Source Data file.

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

## Acknowledgements

This work was supported by Natural Sciences and Engineering Research Council of Canada (NSERC), Canada Research Chair (CRC) Program, Canada Foundation for Innovation (CFI) and the University of Western Ontario. The electron microscopy characterization was carried out at the Canadian Centre for Electron Microscopy (CCEM), a national facility supported by NSERC, the Canada Foundation for Innovation (via the MSI program) and McMaster University. LLM was supported by the Science Challenge Project (TZ2018004), the Fundamental Research Funds for the Central Universities, the National Natural Science Foundation of China (grant no. 51861130360, 51572016, and U1530401), and Newton Advanced Fellowships under the grant No. NAF/R1/180242, the computation supports from Tianhe-2JK computing time award at the Beijing Computational Science Research Center (CSRC).

## Author contributions

L.Z. conceived and designed the experimental work and prepared the manuscript; H.L., Q.W., M.G. and G.B. carried out atomic-resolution STEM tests; R.S. and L.L. performed the DFT calculations; M.N.B., Z.S., J.L. and R.L. helped with ALD characterization and ICP-OES test; N.C., K.A., Z.W., J.C., and T.S. carried out the X-ray absorption fine structure and analysis; X.S. supervised the overall project. All authors have given approval to the final version of the manuscript.

## Competing interests

The authors declare no competing interests.
