## [Peer Review File · Nature Communications]

Reviewers' comments:

Reviewer #1 (Remarks to the Author):

The manuscript focuses on an interesting study on development of Pt single atoms and Pt-Ru dimers together with the identification of active sites for the hydrogen evolution reaction (HER). The manuscript is well-written. The HER performance of the Pt single atom and Pt-Ru dimer catalysts is as good as a Pt/C catalyst. I regret to say however, that there must be some uncertainty in characterization for Pt single atoms and Pt-Ru dimers by STEM and XAS and also interpretations of the HER mechanisms, and I am not fully convinced of their analysis. I consider that the present manuscript will not be suitable to publication to Nature Communications. The following is my comments.

(1) Figure 1: In these illustrations, Pt single atoms are bonded only with N atoms, but in Figure S4, Pt single atoms are bonded to one N atom and two C atoms. Why two different configurations are shown? If Pt single atoms are bonded to one N atom and two C atoms, then the formation of Pt-Ru dimers must be difficult. The author should clarify this point.

(2) Figure 2: From Figure 2b, it is hard to distinguish Pt and Ru atoms by difference in brightness. The authors should present clearer STEM images which can convince readers of a threshold in brightness to distinguish Pt and Ru atoms.

(3) Presumably, ex situ XAS measurements were done for both the Pt single atom and Pt-Ru dimer specimens (without electrochemical controls in a solution). In case, Pt atoms are oxidized and the peaks around 1.6 Å (together with the high WL intensities) may involve a significant Pt-O contribution. However, the authors didn't discuss the presence of Pt-O at all in the manuscript (except for Figure S20, where a PtO₂ spectrum is somehow used for the LCF analysis). A similar argument can be made for Ru-O for the XAS data in Figures 3d and 3e. The authors should discuss the M-O effects on the XAS analysis.

(4) Page 10, lines 183-186: "Detailed examination of the Ru K XANES spectra clearly shows a shift of the edge for Pt-Ru dimers to higher energies compared to Ru metal, which indicates the charge redistribution also in favor of a charge depletion at the Ru site relative to Ru metal". This E₀ shift also can be explained by the presence of Ru-O. The higher unoccupied d state of Pt dimers compared with that of Pt single atoms implies that some d-state electrons in Pt shift to Ru, and this implication may contract to the observation such as the E₀ shift in Ru spectra?

(5) Figure 4: Specify Pt loadings for a Pt/C, Pt single atoms, and Pt dimers.

(6) Page 11, line 207: "When normalized to the metal loading" Specify Pt loadings for a Pt/C, Pt single atoms, and Pt dimers.

(7) Page 11, lines 214 -215: "the Tafel slope of Pt-Ru dimers, Pt single atoms and Pt/C catalysts was 28.9, 33.2 and 29.7 mV dec⁻¹, respectively". It is generally accepted that a first step in the HER in acidic media is the Volmer reaction, in which H is adsorbed (H_{ads}) on a metal (Pt). Then the step is followed by either an electrochemical desorption step (Heyrovsky reaction) or a recombination step (Tafel reaction, such as H_{ads} + H_{ads} → H₂). The Tafel slope for the Tafel reaction is around 30 mV. But the recombination step requires two adjacent Pt sites, and thus the observed Tafel slope of 33.2 mV dec⁻¹ for Pt single atoms doesn't agree with this model. The second step on Pt single atoms must be done through the Heyrovsky reaction. The authors should clarify this point, and also discuss the HER mechanism for Pt-Ru dimers.

(8) Page 18, Table 2: Specify error bars for CNs and bond lengths.

(9) Page 19, Table 3: Specify error bars for CNs and bond lengths.

(10) Page 19, lines 351-352: "Based on LCF qualitative analysis, there is certain amount of Pt-Ru dimer exist in the sample". The percentage of DFT simulated dimer in Figure S20 is low (only around 20%). Specify all the percentages for each constituent for the LCF total (Pt single atoms, PtO₂, and DFT simulated dimer). Also specify the reason why the PtO₂ spectrum was involved.

(11) Page 20, lines 361-362: "XANES features "A", "B" and "C" have all been reproduced". shows that the XANES spectrum in Figure S21 doesn't show the "A" feature at all.

Reviewer #2 (Remarks to the Author):

In this manuscript, the authors presented an effective method for the synthesis of high-quality Pt-Ru dimers on nitrogen-doped carbon nanotubes through atomic layer deposition. The optimized catalysts exhibited superior HER performance in comparison Pt-based counterpart and commercial Pt/C. Significantly, the critical roles of Pt and Ru in electrocatalysis were also identified and the synergistic effect between Pt and Ru play a key role in enhancing HER performance. This model greatly deepen the understanding the catalytic nature and was beneficial to shed light on the structure-activity relationship at an atomic scale. This work presents lots of novelty and significance, and would be of great interest to the related research community. This manuscript is recommended to be accepted after solving the following issues.

1. The existence of PtRu dimer in Fig. 2 should be further characterized because only the observation of the dimer is not sufficient. The composition of the dimer must be verified.
2. As you mentioned, Ru atoms are not effectively attached onto NCNTs during the several initial ALD cycles. The related references were not provided and detailed discussion should be added.
3. Note that the dimer structure was not very uniform within the sample and some nanoclusters were also found. For the EXAFS spectra in Fig. 3C, why Pt-Pt bonding was not observed?
4. No obvious hydrogen adsorption/desorption peak is observed on Pt-Ru dimers and Pt single atoms (Supplementary Fig. 5). However, the electrochemical behaviors are also different due to the existence of redox peak in Fig. S5B. Please explain.
5. As mentioned above, the verification of the existence of dimer (morphology and composition) after stability test should be further carried out. Fig. S11 was not enough.
6. Much noise in Fig. S12 was observed, why? Furthermore, 1000 s was too short.

Reviewer #3 (Remarks to the Author):

The author synthesized atomic layer deposited Pt-Ru dual-metal dimers. I will be mainly focusing on the theoretical side of the work, because of my limited expertise. It is impressive that the Pt-Ru dimers on graphene show much higher mass activity and significantly improved stability compared to commercial Pt/C catalysts for HER. I have several questions for the DFT results, which need be resolved before I can recommend it published in Nat. Commun.

1. Are there any Pt-Pt or Ru-Ru dimers on graphene? I want to know if Pt-Pt has possible activity towards HER? Maybe DFT calculations can answer this question and it might be interesting to compare the results with that for Pt-Ru.
2. I suggest the authors could evaluate the stability of Pt or Ru in Pt-Ru supported by graphene. Once this is done, it can further confirm the stability observed by experimental side.
3. The color scheme (Figure 5) for the atoms should be changed. For example, change red for Ru to other color because it looks like an oxygen.
4. Please evaluate the stability of Pt-Ru dimer when six H atoms adsorb on it. Is it still stable?

5. In Figure 5, the adsorption energies are listed. Is it the average value for H adsorption energy? Please clarify. The authors indicate that "As for one hydrogen atom, it prefers to adsorb on the Ru atom with an adsorption energy of -3.3 eV rather than the Pt atom (-2.5 eV)." Why the adsorption energies for H atoms are only -1.20 eV?

6. "When one or two hydrogen atoms adsorb on Ru or Pt atom, the corresponding free energy is about 0.6 eV." How to obtain this value? Please provide more details. Also for "Pt atom shows the $[\Delta G]_{-H}$ of 1.45eV, and the Ru atom exhibits even low $[\Delta G]_{-H}$ of 0.01 eV". I think Figure 5 confuses me and I strongly suggest the authors should carefully modify Figure 5.

Minor error:

1. In page 14, "Pt or Ru atom could adsorbs" should be "Pt or Ru atom could adsorb".
2. In Figure 5b, in vertical coordinates, ΔG should be ΔG .
3. The structure for Pt(2H)Ru(3H) is not quite clear.

Response to Reviewers' Comments (Manuscript ID NCOMMS-19-14267)

We are very grateful for the reviewers' constructive comments and insightful suggestions for improving the quality of this manuscript. In this revision, we have carried out more experiments, improved 4 figures, and added 6 new figures to give more characterization for the Pt-Ru dimers and to understand the in-depth HER mechanism on the dimers. We also cited more related references and added corresponding discussion in the revised manuscript. The yellow highlights indicate what have been added in the manuscript. The major new experiments, results, and conclusions include:

1. We carried out additional work on atomic resolution STEM images. We obtained further evidence about Pt and Ru information. The quality of Fig. 2 is significantly improved. Now the Pt and Ru atoms in one dimer can be clearly distinguished by the difference in brightness.
2. We carried out more experiments and first-principle calculation to verify the presence of Pt-O. By comparing the XAS spectrum of Pt-Ru dimers with those of PtO₂ and post-testing samples, we found that it is more possible that this structure does not contain Pt-O structure.
3. We did the LCF qualitative analysis again without PtO₂. The percentage of Pt-Ru dimer is clearly specified in the revised manuscript.
4. For the stability of dimers after HER test, we tested the atomic resolution STEM images for the post-testing dimer sample, and the Pt and Ru atoms in one dimer can be clearly distinguished. In addition, the XAS spectrum of Pt and Ru did not change after the stability test, indicating the good stability of dimer structure.
5. As the second reviewer suggested, we carried out a long-term stability test for 10 h. The Pt-Ru dimers and Pt single atoms both exhibited better stability than the Pt/C catalysts.
6. We examined the HER activity of Pt-Pt dimer by DFT calculations, the ΔG_H of Pt-Pt dimer was calculated to be -0.14 eV at the maximal adsorption. This value is very close to that of Pt surface, but it is not as good as that of Pt-Ru dimer. In order to know the stability of the Pt-Ru dimer on the graphene structure, the first-principle molecule dynamics (FPMD) were carried out.

Reviewer: 1

General Comments R1: *The manuscript focuses on an interesting study on development of Pt single atoms and Pt-Ru dimers together with the identification of active sites for the hydrogen evolution reaction (HER). The manuscript is well-written. The HER performance of the Pt single atom and Pt-Ru dimer catalysts is as good as a Pt/C catalyst. I regret to say however, that there must be some uncertainty in characterization for Pt single atoms and Pt-Ru dimers by STEM and XAS and also interpretations of the HER mechanisms, and I am not fully convinced of their analysis. I consider that the present manuscript will not be suitable to publication to Nature Communications. The following is my comments.*

Response: We really appreciate the reviewer's suggestions and questions, which are very helpful to improve the quality of this manuscript. We carried out more characterizations to further verify the formation of Pt-Ru dimers. Firstly, the atomic resolution STEM images are significantly improved. Now the Pt and Ru atoms in one dimer can be clearly distinguished by the difference in brightness. Secondly, we carried out more XAS measurements and first-principle calculations to prove the correctness of the Pt-Ru dimer model. Furthermore, we explained the detailed HER process on Pt single atom and Pt-Ru dimers in the revised manuscript.

Specific Comment R1-1: *Figure 1: In these illustrations, Pt single atoms are bonded only with N atoms, but in Figure S4, Pt single atoms are bonded to one N atom and two C atoms. Why two different configurations are shown? If Pt single atoms are bonded to one N atom and two C atoms, then the formation of Pt-Ru dimers must be difficult. The author should clarify this point.*

Response: We thank the referee for the suggestions. We modified the model to show Pt-C bond in Figure 1. After the deposition of Pt single atoms, Pt is bonded with one N and two C atoms. The most stable structure is predicted by first-principle calculations, which is also shown in Figure S17. It should be pointed that, after the further deposition of Ru, the bonding environment of Pt is changed. As shown in Figure 4, the Ru atom will bond with the two C atoms by breaking the original Pt-C bonding. Meanwhile Pt is pushed outside and only forms the bonding with N atom, except the new bonding with the Ru atom.

ACTION for Fig. 1: We modified the model to show Pt-C bond in Figure 1.

Fig. 1. Schematic illustration of ALD synthesis of Pt-Ru dimers on nitrogen-doped carbon nanotubes (NCNTs). Blue: N, yellow: Pt, red: Ru.

Specific Comment R1-2: Figure 2: From Figure 2b, it is hard to distinguish Pt and Ru atoms by difference in brightness. The authors should present clearer STEM images which can convince readers of a threshold in brightness to distinguish Pt and Ru atoms.

Response: We thank the referee for the suggestions. We carried out additional work on atomic resolution TEM images. We obtained further evidence about Pt and Ru information. The quality of Figure 2b is significantly improved. As shown in Fig. 2b and 2c, the intense of two atoms in one dimer are different, indicating the formation of Pt-Ru dimers. In addition, the Pt and Ru atoms in one dimer can be clearly distinguished by the difference in brightness (Fig. 2d). We have added corresponding description in the revised manuscript.

Page 7:

The two atoms in the dimer shows different contrast (Fig. 2d), which indicates the dimer-like structure is composed of two different elements (in this case are Pt and Ru atoms). In addition, the appearance of the Pt-Ru dimer structure is further confirmed by the XAS results that discussed below.

ACTION for Fig. 2: We obtained two clear atomic resolution STEM images (Fig. 2b and 2c). We identified Pt and Ru atoms in one dimer by the difference in brightness (Fig. 2d).

Fig. 2. Characterizations of Pt-Ru dimers. (a–c) Aberration-corrected HAADF-STEM images of Pt-Ru dimers/NCNTs. The slowly varying contrast is due to the changes in the thickness of the substrate. (d) The intensity profile obtained on one individual Pt-Ru dimer. (e) Distribution histogram showing the ratio between dimers, single atoms and clusters. We determined the percentage of dimers by counting 200 clusters. (f) Pt–Ru distance in the observed Pt-Ru dimers. (g) The ratio of Pt and Ru loading in Pt-Ru dimers/NCNTs, which were obtained under different ALD conditions.

Specific Comment R1-3: Presumably, *ex situ* XAS measurements were done for both the Pt single atom and Pt-Ru dimer specimens (without electrochemical controls in a solution). In case, Pt atoms are oxidized and the peaks around 1.6 Å (together with the high WL intensities) may involve a significant Pt-O contribution. However, the authors didn't discuss the presence of Pt-O at all in the manuscript (except for Figure S20, where a PtO₂ spectrum is somehow used for the LCF analysis). A similar argument can be made for Ru-O for the XAS data in Figures 3d and 3e. The authors should discuss the M-O effects on the XAS analysis.

Response: We thank the referee for the suggestions. As the reviewer suggested, we considered to involve Pt-O in the model. However, it is more possible that this structure does not contain Pt-O structure. The following are our evidence:

(1) We tested the XANES and EXAFS spectrum of PtO₂ (Supplementary Fig. 5). As shown in the Fourier transform spectra from EXAFS of the PtO₂, two obvious peaks can be observed at around 2.7 and 3.3, which is consistent with another work (*J. Am. Chem. Soc.* 2019, 141, 4505-4509). However, in the Pt single atom sample, we did not observe a peak at 2.7 or 3.3 Å. This result indicate that no Pt-O formed in Pt single atoms.

(2) We performed first-principles calculation to examine the Pt-O bonding strength through calculating the O adsorption energy on Pt single atom. The result shows that the O adsorption energy is only 0.27 eV on the Pt single atom, indicating that the Pt-O bond is weak.

(3) We carried out the XAS test for the dimer structures after HER test. If there is Pt-O bond before HER, the spectra should change due to the reduction of the sample. However, the result shows that the spectra is not significantly changed as shown in Supplementary Fig. 13.

(4) In several previous studies, it has been proved that the metal single atoms can be stabilized by bonding with C and N on the substrates (*J. Am. Chem. Soc.* 2019, 141, 4505-4509; *J. Am. Chem. Soc.* 2017, 139, 9419–9422; *J. Am. Chem. Soc.* 2017, 139, 8078–8081; *Angew. Chem. Int. Ed.* 2018, 57, 1944–1948). It is reliable that the Pt single atom and Pt-Ru dimer do not contain Pt-O and Ru-O bond.

We have added corresponding description in the revised manuscript.

Page 9:

The EXAFS R space curve attributes the FT magnitude peak at 1.6 Å to Pt bonded to C, N or O upon which Pt is anchored. To exclude the Pt-O bond, we tested the XANES and EXAFS spectrum of PtO₂ (Supplementary Fig. 5). As shown in the Fourier transform spectra from EXAFS of the PtO₂, two obvious peaks can be observed at around 2.7 and 3.3 Å, respectively. However, in the Pt single atom sample, we did not observe a peak at 2.7 either 3.3 Å (Fig. 3c). This result indicates that no Pt-O formed in Pt single atoms. In addition, previous studies showed that the metal single atoms can be stabilized by bonding with C and N on the substrates.⁴⁰⁻⁴³ Thus, we concluded that the peak at 1.6 Å is attributed to Pt-N/Pt-C bond.

Page 12:

In addition, the TEM images of the post-testing Pt-Ru dimer catalysts indicated that the dimer structure is stable after the durability test (Supplementary Fig. 12). The different intense of two atoms in one dimer indicate the existence of dimer structure. The mass ratio of Pt and Ru in the post-testing Pt-Ru dimer catalysts was still around 3.0, which is determined by ICP-OES. We also carried out the XAS test for the dimer structures after HER test. The XANES spectrum of Pt has almost no change. In the K₂-weighted magnitude of Fourier transform spectra from EXAFS spectrum, a small peak at 2.8 Å attributed to the Pt-Ru scattering can still be observed

(Supplementary Fig. 13). For the Ru XAS spectra, three XANES features “A”, “B” and “C” can be reproduced, indicating the maintenance of the Pt-Ru dimer structure (Supplementary Fig. 14).

Specific Comment R1-4: *Page 10, lines 183-186: “Detailed examination of the Ru K XANES spectra clearly shows a shift of the edge for Pt-Ru dimers to higher energies compared to Ru metal, which indicates the charge redistribution also in favor of a charge depletion at the Ru site relative to Ru metal”. This E_0 shift also can be explained by the presence of Ru-O. The higher unoccupied d state of Pt dimers compared with that of Pt single atoms implies that some d-state electrons in Pt shift to Ru, and this implication may contract to the observation such as the E_0 shift in Ru spectra?*

Response: We appreciate the referee for the comments. (1) About the presence of Ru-O. To investigate the presence of Ru-O, we carried out the Ru XANES test after HER test. If there is Ru-O bond before HER, the spectra should change due to the reduction of the sample. However, the result shows that the spectra is almost not changed as shown in Supplementary Figure S14. Similarly as Pt, we concluded that no Ru-O formed during our preparation process. (2) About the electron transfer between Pt and Ru. Indeed, the bonding between Pt and Ru can affect their electronic structures. However, positive E_0 shift of Ru is not due to the d-state electrons transfer from Pt. The main reason caused the E_0 shift in Ru spectra was due to the bonding between Ru and C. In the DFT part on page 16, we investigated the partially density of states for both the Pt and Ru atoms. The result shows that both Pt atom and Ru atom of dimer exhibits many unoccupied states, which is due to the strong Ru-C and Pt-N bonds. We have added corresponding description in the revised manuscript.

Page 10:

The positive E_0 shift of Ru is due to the bonding between Ru atoms and C on the substrates.

Page 12:

For the Ru XAS spectra, three XANES features “A”, “B” and “C” can be reproduced, indicating the maintenance of the Pt-Ru dimer structure (Supplementary Fig. 14).

Page 18:

As shown in Supplementary Fig.19, before the hydrogen adsorption, both Pt atom and Ru atom of dimer exhibits unoccupied states, which should origin from the bonding between metal ion and C/N atoms.

Specific Comment R1-5: *Figure 4: Specify Pt loadings for a Pt/C, Pt single atoms, and Pt dimers.*

Response: We appreciate the referee for the comments. We have added the loading amount of these three catalysts in the caption of Figure 4.

Specific Comment R1-6: Page 11, line 207: “When normalized to the metal loading” Specify Pt loadings for a Pt/C, Pt single atoms, and Pt dimers.

Response: We appreciate the referee for the comments. We have added the loading amount of these three catalysts in the revised manuscript.

Page 11:

The metal mass loadings for Pt/C, Pt single atoms, and Pt dimers are 40.0%, 0.9% and 1.2%, respectively.

Specific Comment R1-7: Page 11, lines 214 -215: “the Tafel slope of Pt-Ru dimers, Pt single atoms and Pt/C catalysts was 28.9, 33.2 and 29.7 mV dec⁻¹, respectively”. It is generally accepted that a first step in the HER in acidic media is the Volmer reaction, in which H is adsorbed (H_{ads}) on a metal (Pt). Then the step is followed by either an electrochemical desorption step (Heyrovsky reaction) or a recombination step (Tafel reaction, such as $H_{ads} + H_{ads} \rightarrow H_2$). The Tafel slope for the Tafel reaction is around 30 mV. But the recombination step requires two adjacent Pt sites, and thus the observed Tafel slope of 33.2 mV dec⁻¹ for Pt single atoms doesn't agree with this model. The second step on Pt single atoms must be done through the Heyrovsky reaction. The authors should clarify this point, and also discuss the HER mechanism for Pt-Ru dimers.

Response: We appreciate the referee for the comments. (1) The HER process on Pt particles contains two half reactions, which are the Volmer reaction and the desorption reaction (Heyrovsky reaction or Tafel reaction). For the Pt single atom catalysts, it has been reported that several H could adsorb on one Pt atom, followed by the formation of H₂ on the isolated Pt atom (Nat. Commun. 2016, 7, 13638; Nat. Commun. 2017, 8, 1490; ACS Catal. 2018, 8, 8450-8458). The desorption process is similar as the Tafel reaction ($H_{ads} + H_{ads} \rightarrow H_2$), which results in the similar Tafel slope. (2) For the HER mechanism on Pt-Ru dimers, we carefully explained the reaction route by first-principle calculations. We explored how the H adsorption evolves on the Pt-Ru dimer step by step, and the different numbers of H adsorption on the Pt-Ru dimer were examined (Figure 5). We found that when three hydrogen atoms connect with the Pt atom (3H-Pt), and three bonds with Ru atom (3H-Ru), the hydrogen atom becomes easy to detach from Ru atom. We have added corresponding description in the revised manuscript.

Page 12:

It should be noted that the Pt SA is different from the Pt surface, and the former could adsorb more than one hydrogen atom on the one isolated atom, and each Pt atom of the later typically adsorbs one hydrogen atom at the surface^{18,30,31}. Thus the Pt SA can process the HER process on the isolated Pt atom through the Tafel half reaction ($H_{ads} + H_{ads} \rightarrow H_2$).

Page 16:

The calculated hydrogen adsorption Gibbs free energies (ΔG_H) of the Pt-Ru dimer were performed to examine the activity of the HER for different number of H atoms, and the results

for typical configurations are shown in Fig. 5b. When one hydrogen atom adsorbs on both Ru and Pt sides of the Pt-Ru dimer, the corresponding Gibbs free energy for Pt(1H)Ru(1H) is about -1 eV. When another hydrogen adsorbs on the dimer, the corresponding Gibbs free energy for Pt(2H)Ru(1H) becomes about 0.6 eV. This means that the hydrogen atoms are anchored on Ru atom steadily or hard to detach from either Ru or Pt atom at this stage. As for the last step (see Fig. 5), three hydrogen atoms connect with the Pt atom (3H-Pt), and three bonds with Ru atom (3H-Ru). The Ru atom exhibits the low ΔG_H of 0.01 eV through the pathway of Pt(3H)Ru(3H) \rightarrow Pt(3H)Ru(2H), which clearly indicates that the hydrogen atom becomes easy to detach from Ru when the maximum coverage of 6H is reached.

Specific Comment R1-8: *Page 18, Table 2: Specify error bars for CNs and bond lengths.*

Response: We thanks the referee for the comments. We have added the specify error bars in Table 2.

Specific Comment R1-9: *Page 19, Table 3: Specify error bars for CNs and bond lengths.*

Response: We thanks the referee for the comments. We have added the specify error bars in Table 3.

Specific Comment R1-10: *Page 19, lines 351-352: “Based on LCF qualitative analysis, there is certain amount of Pt-Ru dimer exist in the sample”. The percentage of DFT simulated dimer in Figure S20 is low (only around 20%). Specify all the percentages for each constituent for the LCF total (Pt single atoms, PtO₂, and DFT simulated dimer). Also specify the reason why the PtO₂ spectrum was involved.*

Response: We appreciate the referee for the comments. In our revised manuscript, we concluded that there were no PtO₂ involved in this sample. Therefore, we did the LCF qualitative analysis again without PtO₂. The result show that the percentage of Pt-Ru dimer and Pt single atoms are 15% and 85%, respectively. Because there are some Pt clusters in this sample, the amount of Pt single atom is overestimated. However, this data can prove the existence of certain amount Pt-Ru dimer in the sample. We have added the corresponding discussion in the revised manuscript.

Page 20:

The result show that the percentage of Pt-Ru dimer and Pt single atoms are 15% and 85%, respectively. Because there are some Pt clusters in this sample, the amount of Pt single atom is overestimated. However, this data can prove the existence of certain amount Pt-Ru dimer in the sample.

Specific Comment R1-11: *Page 20, lines 361-362: “XANES features “A”, “B” and “C” have all been reproduced”. shows that the XANES spectrum in Figure S21 doesn’t show the “A” feature at all.*

Response: We appreciate the referee for the comments. As shown in Figure S21, the XANES fitting for Ru with different cluster radius showed feature A. Because R5.0 shows a relatively strong B feature, the feature A was almost overlapped. The feature A appears as a shoulder peak. It should be pointed that the peak intense and peak position of the fitted data might be a little different with the tested data, because the model might have some defects compared with the real sample. However, the three XANES features “A”, “B” and “C” can still be used as the evidence of the formation of Pt-Ru dimers.

Page 21:

The spectrum of Ru XANES modeling with the Ru radius of 5.0 Å shows a relatively strong B feature and a shoulder peak for A feature. It should be pointed that the peak intense and peak position of the fitted data might be a little different with the experimental data, because the model might have some defects compared with the actual Ru local structural environment of the sample. The presence of the three XANES features “A”, “B” and “C” can be used as the evidence of the formation of Pt-Ru dimers.

Reviewer 2:

General Comments R2: *In this manuscript, the authors presented an effective method for the synthesis of high-quality Pt-Ru dimers on nitrogen-doped carbon nanotubes through atomic layer deposition. The optimized catalysts exhibited superior HER performance in comparison Pt-based counterpart and commercial Pt/C. Significantly, the critical roles of Pt and Ru in electrocatalysis were also identified and the synergistic effect between Pt and Ru play a key role in enhancing HER performance. This model greatly deepen the understanding the catalytic nature and was beneficial to shed light on the structure-activity relationship at an atomic scale. This work presents lots of novelty and significance, and would be of great interest to the related research community. This manuscript is recommended to be accepted after solving the following issues.*

Response: We appreciate the referee for the kind comments.

Specific Comment R2-1: *The existence of PtRu dimer in Fig. 2 should be further characterized because only the observation of the dimer is not sufficient. The composition of the dimer must be verified.*

Response: We thank the referee for the suggestions. We tested the atomic resolution TEM images again. We obtained further evidences. The quality of Figure 2b is improved. As shown in Fig. 2b and 2c, the intense of two atoms in one dimer are different, indicating the formation of Pt-Ru dimers. In addition, the Pt and Ru atoms in one dimer can be clearly distinguished by the difference in brightness (Fig. 2d). We have added corresponding description in the revised manuscript.

Page 7:

The two atoms in the dimer shows different contrast (Fig. 2d), which indicates the dimer-like structure is composed of two different elements (in this case are Pt and Ru atoms). In addition, the appearance of the Pt-Ru dimer structure is further confirmed by the XAS results that discussed below.

ACTION for Fig. 2: We obtained two clear atomic resolution STEM images (Fig. 2b and 2c). We identified Pt and Ru atoms in one dimer by the difference in brightness (Fig. 2d).

Fig. 2. Characterizations of Pt-Ru dimers. (a–c) Aberration-corrected HAADF-STEM images of Pt-Ru dimers/NCNTs. The slowly varying contrast is due to the changes in the thickness of the substrate. (d) The intensity profile obtained on one individual Pt-Ru dimer. (e) Distribution histogram showing the ratio between dimers, single atoms and clusters. We determined the percentage of dimers by counting 200 clusters. (f) Pt–Ru distance in the observed Pt-Ru dimers. (g) The ratio of Pt and Ru loading in Pt-Ru dimers/NCNTs, which were obtained under different ALD conditions.

Specific Comment R2-2: *As you mentioned, Ru atoms are not effectively attached onto NCNTs during the several initial ALD cycles. The related references were not provided and detailed discussion should be added.*

Response: We appreciate the referee for the comments. We have added our previous reference and detailed discussion in the revised manuscript.

Page 6:

In our previous study, we found that Ru atoms can not be deposited on NCNTs in the first several ALD cycles.³⁶ This result indicates that Ru atoms are not effectively attached onto

NCNTs during the several initial ALD cycles, which provided the necessary prerequisite for selectively deposition of Ru atoms on Pt atoms.

Page 31:

36. Zhang, L. et al. Selective atomic layer deposition of RuOx catalysts on shape-controlled Pd nanocrystals with significantly enhanced hydrogen evolution activity. *J. Mater. Chem. A* **6**, 24397-24406 (2018).

Specific Comment R2-3: *Note that the dimer structure was not very uniform within the sample and some nanoclusters were also found. For the EXAFS spectra in Fig. 3C, why Pt-Pt bonding was not observed?*

Response: We appreciate the referee for the comments. As around 90% of the atoms are single atoms in the Pt single atom sample, the Pt-Pt peak (2.6 Å) is very weak. As shown in Fig. 3c, there is a very weak shoulder peak for Pt single atoms. After the deposition of Ru, the peak at 2.6 Å become much obvious, indicating the formation of Pt-Ru bond.

Specific Comment R2-4: *No obvious hydrogen adsorption/desorption peak is observed on Pt-Ru dimers and Pt single atoms (Supplementary Fig. 5). However, the electrochemical behaviors are also different due to the existence of redox peak in Fig. S5B. Please explain.*

Response: We appreciate the referee for the comments. The existence of redox peak in Figure S5B is much weak compared with the regular hydrogen adsorption/desorption on Pt/C. Pt single atoms did not show any peaks. However, Ru in the dimer structure can adsorb O easily and proceed an oxidation–reduction process than Pt. As a result, Fig. S5b exhibited a weak O adsorption/desorption peak. We have added relevant description in the revised manuscript.

Page 10:

As Ru in the dimer structure can adsorb and desorb O more easily through an oxidation–reduction process than Pt, the CV curve in Supplementary Fig. 6b exhibited a weak O adsorption/desorption peak.

Specific Comment R2-5: *As mentioned above, the verification of the existence of dimer (morphology and composition) after stability test should be further carried out. Fig. S11 was not enough.*

Response: We appreciate the referee for the comments. We tested the atomic resolution STEM images for the post-testing dimer sample again. The Pt and Ru atoms in one dimer can be clearly distinguished by the difference in brightness (Supplementary Fig. 12). In addition, we obtained the XAS spectrum for the post-testing dimer sample. The electronic structure of Pt and Ru did

not changed after the stability test, indicating the good stability of dimer structure. We have added corresponding description in the revised manuscript.

Page 12:

In addition, the STEM images of the post-testing Pt-Ru dimer catalysts indicated that the dimer structure is stable after the durability test (Supplementary Fig. 12). The different intense of two atoms in one dimer indicate the existence of dimer structure. We also carried out the XAS test for the dimer structures after HER test. The XANES spectrum of Pt has almost no change. In the K₂-weighted magnitude of Fourier transform spectra from EXAFS spectrum, a small peak at 2.8 Å attributed to the Pt-Ru scattering can still be observed (Supplementary Fig. 13). For the Ru XAS spectra, three XANES features “A”, “B” and “C” can be reproduced, indicating the maintenance of the Pt-Ru dimer structure (Supplementary Fig. 14).

Specific Comment R2-6: *Much noise in Fig. S12 was observed, why? Furthermore, 1000 s was too short.*

Response: We appreciate the referee for the comments. We have carried out a long-term stability test for 10 h. The Pt-Ru dimers and Pt single atoms both exhibited better stability than the Pt/C catalysts. We have added corresponding description in the revised manuscript.

Page 12:

We also examined the long-term stability of the Pt-Ru dimers, Pt single atoms and commercial Pt/C by extended electrolysis at fixed current density of 10 mA cm⁻² for 10 h. Supplementary Fig. 15 shows that the Pt-Ru dimers and Pt single atoms exhibit significantly higher HER stability compared to the commercial Pt/C. Compared to the 95.1 mV potential drop for the Pt/C, there are only 19.6 and 56.9 mV potential drop for Pt-Ru dimers and Pt single atoms, respectively. Due to the strong interaction between Pt atoms with the N-site, both Pt-Ru dimers and Pt single atoms show better stability than the Pt/C catalysts.

Reviewer 3:

General Comments R3: *The author synthesized atomic layer deposited Pt-Ru dual-metal dimers. I will be mainly focusing on the theoretical side of the work, because of my limited expertise. It is impressive that the Pt-Ru dimers on graphene show much higher mass activity and significantly improved stability compared to commercial Pt/C catalysts for HER. I have several questions for the DFT results, which need be resolved before I can recommend it published in Nat. Commun.*

Response: We appreciate the referee for the kind comments.

Specific Comment R3-1: *Are there any Pt-Pt or Ru-Ru dimers on graphene? I want to know if Pt-Pt has possible activity towards HER? Maybe DFT calculations can answer this question and it might be interesting to compare the results with that for Pt-Ru.*

Response: We appreciate the referee for the comments. (1) About the existence of Pt-Pt or Ru-Ru dimers. As shown in our experimental side, the preparation of the dimer was separated into two steps. The Pt single atoms were firstly prepared, and then the Ru atoms were further deposited in the second stage. As shown in the HAADF-STEM images, no Pt-Pt dimers were observed after the first ALD step (see Figure S1-S2). The dimers were only observed in the second stage. Meanwhile as shown in our updated HR-STEM images (Fig. 2b and 2c), the different bright spots correspond to the Ru and Pt, respectively, which further proved the formation of Pt-Ru dimer instead of Ru-Ru dimer. On the other side, after the deposition of Pt single atoms, most of the defect sites of NCNTs were occupied. The calculated adsorption energies of Ru dimer on the perfect graphene structure and Pt-Ru dimer on the N-doped site are -1.3 eV and -4.87 eV, which indicates that the Ru prefers to form the Pt-Ru dimer instead of Ru-Ru dimer. (2) About the HER activity of Pt-Pt dimers. In order to test the HER of Pt-Pt dimer, the ΔG_H of Pt-Pt dimer was calculated compared with that of Pt-Ru dimer. The results show that Pt-Pt dimer can also adsorb the maximal 6H on the Pt-Pt dimer, and the calculated ΔG_H is -0.14 eV at the maximal adsorption. This value is very close to that of Pt surface (Nat. Commun. 2016, 7, 13638), but it is not as good as that of Pt-Ru dimer, as discussed in the context. We have added corresponding description in the revised manuscript.

Page 15:

Considering the Ru atoms were deposited after the Pt SAs were formed, most of the N-sites have been occupied by the Pt atoms. Thus the Ru atoms could only either bond with Pt to form Pt-Ru dimer or form Ru-Ru dimer. The adsorption energies of Ru-Ru dimer on the perfect site and Pt-Ru dimer on the N-doped site were calculated to be -1.3 eV and -4.87 eV, which indicates that the Ru prefers to form the Pt-Ru dimer instead of Ru-Ru dimer. In addition, as we clearly identify the Pt-Ru atomic pairs from HR-TEM images and Pt-Ru bonding from EXAFS spectra, the following first-principle calculations were carried out using one Pt-Ru dimer on the N-doped graphene to identify reactivity of the dimer.

Page 16:

Meanwhile, the calculated ΔG_H of Pt-Pt dimer is about -0.14 eV (Supplementary Fig. 18), which is quite close to that the bulk one, inferior to that of Pt-Ru as well.

Specific Comment R3-2: *I suggest the authors could evaluate the stability of Pt or Ru in Pt-Ru supported by graphene. Once this is done, it can further confirm the stability observed by experimental side.*

Response: We appreciate the referee for the comments and very good suggestions. In order to know the stability of the Pt-Ru dimer on the graphene, the first-principle molecule dynamics (FPMD) were carried out. The simulations were carried out with the target temperature of 300 K. The FPMD were carried out for 5 ps. During the whole FPMD, the Pt-Ru dimer was adsorbed on the N-doped graphene, and no big structure change was observed. This result further confirms the high stability of Pt-Ru dimer as observed in experimental side. We have added corresponding description in the revised manuscript.

Page 16:

In order to understand the stability of the pure Pt-Ru and Pt-Ru with 6H on the N-doped graphene, the first principles molecular dynamics (FPMD) were carried out for 5ps with the target temperature of 300 K. The results show that both Pt-Ru dimer with and without the six H were stable, and no big structure change occurred during this process, which further confirm the stability of the Pt-Ru dimer (Supplementary Fig. 16).

Specific Comment R3-3: *The color scheme (Figure 5) for the atoms should be changed. For example, change red for Ru to other color because it looks like an oxygen.*

Response: We appreciate the referee for the comments. We have changed their colors.

Specific Comment R3-4: *Please evaluate the stability of Pt-Ru dimer when six H atoms adsorb on it. Is it still stable?*

Response: We appreciate the referee for the comments. We have carried out the FPMD simulations for both the pure Pt-Ru dimer and Pt-Ru with 6H for 5 ps, and both of systems were stable, which further confirm the stabilities of both systems. As for the details, please refer to the Reply **R3-2** of the same referee.

Specific Comment R3-5: *In Figure 5, the adsorption energies are listed. Is it the average value for H adsorption energy? Please clarify. The authors indicate that “As for one hydrogen atom, it prefers to adsorb on the Ru atom with an adsorption energy of -3.3 eV rather than the Pt atom (-2.5 eV).” Why the adsorption energies for H atoms are only -1.20 eV?*

Response: We appreciate the referee for the comments and thanks for the good suggestion. In Fig. 5a, the adsorption energies are the average value for H adsorption energy, calculated by following equation: $E_{ad} = 1/n(E_{nH/metallic\ dimer} - E_{metallic\ dimer} - n/2 E_{H_2})$. When two hydrogen atoms adsorb on Pt-Ru dimer, the adsorption energies of Pt(1H)Ru(1H) becomes -1.2 eV per hydrogen. The total adsorption energy for Pt(1H)Ru(1H) are even smaller than that of Pt(0H)Ru(1H), which indicates that the second H is not easy to adsorb on the Pt compared with Ru. It should be noted that the definition of the adsorption energy also affects the relative value, which contains the formation of H₂. With the increase the number of H atoms from 4 to 5, the total adsorption energies becomes more negative. The main reason should be related with the transition from metallic to semiconductor for Pt-Ru dimer during this process. We have added corresponding description in the revised manuscript.

Page 15:

It should be noted that the second H is not so easy to adsorb on the Pt-Ru dimer compared with the first one, while with the further increase the number of the hydrogen atoms, the H adsorption becomes favorable again, which should be related with the change of electronic structure for Pt-Ru dimer.

Specific Comment R3-6: *“When one or two hydrogen atoms adsorb on Ru or Pt atom, the corresponding free energy is about 0.6 eV.” How to obtain this value? Please provide more details. Also for “Pt atom shows the $[\Delta G]_H$ of 1.45eV, and the Ru atom exhibits even low $[\Delta G]_H$ of 0.01 eV”. I think Figure 5 confuses me and I strongly suggest the authors should carefully modify Figure 5.*

Response: We appreciate the referee for the comments. First of all, the free energy was calculated by the following two equations: (1) $\Delta E_H = E_{(n+1)H/metal} - E_{nH/metal} - 1/2 E_{H_2}$; (2) $\Delta G_H = \Delta E_H + \Delta E_{ZPE} - T\Delta S$. To clearly state how we obtained the value, we added more details on page 16. Secondly, in order to make Fig. 5 clear, the lowest ΔG_H for each coverage was typically shown in Fig. 5b in the revised manuscript.

Page 16:

The calculated hydrogen adsorption Gibbs free energies (ΔG_H) of the Pt-Ru dimer were performed to examine the activity of the HER for different number of H atoms, and the results for typical configurations are shown in Fig. 5b. When one hydrogen atom adsorbs on both Ru and Pt sides of the Pt-Ru dimer, the corresponding Gibbs free energy for Pt(1H)Ru(1H) is about -1 eV. When another hydrogen adsorbs on the dimer, the corresponding Gibbs free energy for Pt(2H)Ru(1H) becomes about 0.6 eV. This means that the hydrogen atoms are anchored on Ru

atom steadily or hard to detach from either Ru or Pt atom at this stage. As for the last step (see Fig. 5), three hydrogen atoms connect with the Pt atom (3H-Pt), and three bonds with Ru atom (3H-Ru). The Ru atom exhibits the low ΔG_H of 0.01 eV through the pathway of Pt(3H)Ru(3H) \rightarrow Pt(3H)Ru(2H), which clearly indicates that the hydrogen atom becomes easy to detach from Ru when the maximum coverage of 6H is reached.

Specific Comment R3-7: *In page 14, “Pt or Ru atom could adsorbs” should be “Pt or Ru atom could adsorb”.*

Response: We appreciate the referee for the comments. We have corrected it.

Specific Comment R3-8: *In Figure 5b, in vertical coordinates, ΔG should be delta G.*

Response: We appreciate the referee for the comments. We have corrected it.

Specific Comment R3-9: *The structure for Pt(2H)Ru(3H) is not quite clear.*

Response: We appreciate the referee for the comments. We have made it clear.

Many thanks again!

REVIEWERS' COMMENTS:

Reviewer #1 (Remarks to the Author):

The rebuttals from the authors are satisfactory, except for the following technical point. I consider that the manuscript is now suitable for publication to Nature Communications after the following revision.

- The metal loadings for Pt/C, Pt single atoms, and Pt dimers (Fig. 4) should be given by a form of $\mu\text{g}/\text{cm}^2$.

Reviewer #2 (Remarks to the Author):

The authors have revised the manuscript carefully according to reviewers' comments. The presence of the Pt-Ru dimer, the coordination of metal species and their further electrochemical performance have been further verified. This work definitely provides new insight into the rational design of atomically dispersed catalysts and in-depth understanding of the catalytic nature at the atom scale. Overall, this revised manuscript is recommended to be accepted at the current stage.

Reviewer #3 (Remarks to the Author):

I think the responses are satisfactory and it can be accepted in the current state.

Response to Reviewers' Comments (Manuscript ID NCOMMS-19-14267A)

Reviewer: 1

Comment R1: *The rebuttals from the authors are satisfactory, except for the following technical point. I consider that the manuscript is now suitable for publication to Nature Communications after the following revision.*

- *The metal loadings for Pt/C, Pt single atoms, and Pt dimers (Fig. 4) should be given by a form of $\mu\text{g}/\text{cm}^2$.*

Response: We really appreciate the reviewer's suggestions. We have revised the metal loadings by a form of $\mu\text{g}/\text{cm}^2$.

Page 9:

The metal mass loadings for Pt/C, Pt single atoms, and Pt-Ru dimers on the electrode are 61.2, 1.34 and $1.67 \mu\text{g cm}^{-2}$, respectively

Reviewer 2:

General Comments R2: *The authors have revised the manuscript carefully according to reviewers' comments. The presence of the Pt-Ru dimer, the coordination of metal species and their further electrochemical performance have been further verified. This work definitely provide new insight into the rational design of atomically dispersed catalysts and in-depth understanding of the catalytic nature at the atom scale. Overall, this revised manuscript is recommended to be accepted at the current stage.*

Response: We appreciate the referee for the kind comments.

Reviewer 3:

General Comments R3: *I think the responses are satisfactory and it can be accepted in the current state.*

Response: We appreciate the referee for the kind comments.

Many thanks again!